# Frequentist Consistency of Prior-Data Fitted Networks for Causal Inference

**Valentyn Melnychuk** [1]  **Vahid Balazadeh** [2]  **Stefan Feuerriegel** [1]  **Rahul G. Krishnan** [2]

## Abstract

Foundation models based on prior-data fitted networks (PFNs) have shown strong empirical performance in causal inference by framing the task as an in-context learning problem. However, it is unclear whether PFN-based causal estimators provide uncertainty quantification that is consistent with classical frequentist estimators. In this work, we address this gap by analyzing the frequentist consistency of PFN-based estimators for the average treatment effect (ATE). (1) We show that existing PFNs, when interpreted as Bayesian ATE estimators, can exhibit prior-induced confounding bias: the prior is not asymptotically overwritten by data, which, in turn, prevents frequentist consistency. (2) As a remedy, we suggest employing a calibration procedure based on a one-step posterior correction (OSPC). We show that the OSPC helps to restore frequentist consistency and can yield a semi-parametric Bernstein–von Mises theorem for calibrated PFNs (i.e., both the calibrated PFN-based estimators and the classical semi-parametric efficient estimators converge in distribution with growing data size). (3) Finally, we implement OSPC through tailoring martingale posteriors on top of the PFNs. In this way, we are able to recover functional nuisance posteriors from PFNs, required by the OSPC. In multiple (semi-)synthetic experiments, PFNs calibrated with our martingale posterior OSPC produce ATE uncertainty that (i) asymptotically matches frequentist uncertainty and (ii) is well calibrated in finite samples in comparison to other Bayesian ATE estimators.

[1]LMU Munich & Munich Center for Machine Learning (MCML), Munich, Germany [2]University of Toronto & Vector Institute, Toronto, Canada. Correspondence to: Valentyn Melnychuk <melnychuk@lmu.de>.

*Proceedings of the 43rd International Conference on Machine Learning*, Seoul, South Korea. PMLR 306, 2026. Copyright 2026 by the author(s).

## 1. Introduction

Estimating the effect of treatments from observational data is widely relevant for decision-making in marketing (Langen & Huber, 2023), public policy (Lechner, 2023; Kuzmanovic et al., 2024), and medicine (Feuerriegel et al., 2024). A key target estimand here is the *average treatment effect (ATE)*, which quantifies the population-level causal effect of an intervention.

Foundation models based on prior-data fitted networks (PFNs) have recently shown strong empirical performance in causal inference. Under their paradigm, causal inference is framed as an in-context learning problem (Brown et al., 2020; Dong et al., 2024): the estimator/predictor is simply provided by a single forward pass through a large pretrained network (e. g., a transformer) that takes the whole observational dataset as an input.

At a high level, PFNs are trained exclusively on synthetic datasets sampled from a prior over data-generating processes, which thus builds upon ideas of amortized variational Bayesian methods (Garnelo et al., 2018; Kim et al., 2019; Xie et al., 2022). A prominent example is TabPFN (Hollmann et al., 2023; 2025) that is tailored to tabular data (Erickson et al., 2026). Recently, several works adapted the PFNs specifically to causal inference (Robertson et al., 2025; Balazadeh et al., 2025; Ma et al., 2026) (see Table 1).

However, it is unclear whether existing PFNs for causal inference provide reliable uncertainty quantification. Notably, PFNs are approximate Bayesian models, which offer total uncertainty quantification "out-of-the-box" in the form of the *posterior predictive density (PPD)*. Several works have explored how the PPDs can be used for downstream tasks (Jesson et al., 2025; Nagler & Rügamer, 2025). This feature makes PFNs attractive for causal inference (Balazadeh et al., 2025; Ma et al., 2026), since PPDs can yield the uncertainty for both the outcome model and propensity model (i.e., the nuisance functions). Yet, to the best of our knowledge, no work has so far studied PFNs for causal inference in terms of their *frequentist consistency* (i. e., whether PFN-based estimators asymptotically converge to classical semi-parametric frequentist estimators). Our paper aims to fill this gap.

Here, we *study the frequentist consistency of PFNs for ATE estimation*. While we focus on the ATE, our results naturally

*Table 1.* Overview of prior-data fitted networks (PFNs) that are suitable for ATE estimation.

| PFN | POF | ① PICB | Posterior distribution, $\mathbb{P}(\diamond \mid \mathcal{D}, x)$ | | Posterior means | | | | | ② Viable for the OSPC | ③ Viable for MPs |
|---|---|---|---|---|---|---|---|---|---|---|---|
| | | | $Y[a]$ | $Y[1] - Y[0]$ | $\mu_a$ | $A$ | $\mu_a$ | $\pi$ | $\tau$ | | |
| TabPFN (Hollmann et al., 2023) | ✓† | ⚠ | ✓ | ✗ | ✓* | ✓ | ✓ | ✓ | ✓ | ✓ | ✓ |
| Do-PFN (Robertson et al., 2025) | ✗ | ⚠‡ | ✓ | ✗ | ✗‡ | ✗ | ✓ | ✗ | ✓ | ✗ | ✗‡ |
| CausalPFN (Balazadeh et al., 2025) | ✓ | ⚠ | ✗ | ✗ | ✓ | ✗ | ✓ | ✗ | ✓ | ✗ | ✗× |
| CausalFM (Ma et al., 2026) | ✓ | ⚠ | ✗ | ✓ | ✗ | ✗ | ✗ | ✗ | ✓ | ✗ | ✗× |

POF: potential outcomes framework; PICB: prior-induced confounding bias; MPs: martingale posteriors; OSPC: one-step posterior correction
†: when used as a S- or T-learner; ‡: struggles with non-identifiability; *: can be recovered through MPs; ×: do not provide a full PPD.

extend to other finite-dimensional causal estimands. Overall, our paper makes the following contributions:

① (→ Sec. 5.1): First, we show that the existing PFNs (Hollmann et al., 2023; 2025; Balazadeh et al., 2025; Ma et al., 2026), when used as Bayesian ATE estimators, are prone to a so-called *prior-induced confounding bias*.[1] Informally, this bias arises because the PFN's implicit prior (which is learned from synthetic training distributions) can systematically shrink the degree of observed confounding toward zero. As a result, the PFN's posterior is concentrated on nearly unconfounded data-generating processes, so the ATE estimates can be biased even as the sample size grows, which thus prevents the frequentist consistency.

② (→ Sec. 5.2): To address the prior-induced confounding bias in PFNs, we employ a novel calibration procedure based on the efficient influence functions, namely a *one-step posterior correction (OSPC)* (Yiu et al., 2025). The OSPC allows us to recalibrate the PFN's uncertainty without full re-training and, under mild assumptions, restores frequentist consistency. As our main contribution, we show theoretically that a *semi-parametric Bernstein–von Mises (BvM) theorem* (Ray & van der Vaart, 2020) holds approximately for the calibrated PFNs. This has the following implications: the OSPC of the PFNs (i) yields ATE posteriors that asymptotically match the normal distribution given by a frequentist augmented inverse probability of treatment weighted (A-IPTW) estimators, and thereby (ii) restores frequentist consistency.

③ (→ Sec. 5.3): Finally, we implement our OSPC by adapting the general framework of martingale posteriors (MPs) (Fong et al., 2023) to our setting, which we call **MP-OSPC** hereafter. This step is necessary because the OSPC requires not just the PPDs provided by the PFNs but *the ability to sample posteriors over the nuisance functions*. Fortunately, MPs make it possible to recover posteriors for any parameter/function of the underlying data-generating process, as long as the PPD can be sequentially updated (that is, it satisfies a martingale property). To implement our *MP-OSPC*, we tailor a copula-based MP framework (Nagler & Rügamer, 2025) to our causal setting. By calibrating the uncertainty of

the PFNs in a manner targeted specifically at the ATE, our *MP-OSPC* combines the best of both worlds: (i) asymptotic consistency of frequentist estimators and (ii) finite-sample uncertainty guided by the prior of Bayesian estimators. Empirically, we show that existing PFNs combined with our *MP-OSPC* (i) asymptotically match the well-established A-IPTW estimators; and (ii) often achieve better finite-sample calibration compared to other Bayesian ATE estimators.

In sum, our paper is **novel** in three ways:[2] **(1)** We show that naïve PFN-based Bayesian ATE estimators can be systematically biased due to an overly strong implicit prior that is not overwritten by the observed data. **(2)** We develop a novel calibration procedure based on a one-step posterior correction and martingale posteriors, which we call *MP-OSPC*. **(3)** We show empirically that existing PFNs with our *MP-OSPC* on top achieve state-of-the-art uncertainty quantification when compared to standard PFN-based baselines.

## 2. Related Work

### 2.1. Causal Inference

**Frequentist estimators.** A broad family of estimators has been developed for causal inference from observational data, including plug-in and inverse probability treatment weighted (IPTW) estimators, which all require the estimation of nuisance functions (Kennedy, 2024). Among these, an A-IPTW estimator is particularly attractive: it yields semi-parametric $\sqrt{n}$-consistent, efficient, and asymptotically normal inference for finite-dimensional targets (such as ATE) under standard regularity conditions (Robins, 2000; Newey, 1994); and it is, in some sense, optimal (Jin & Syrgkanis, 2025). Conceptually, the A-IPTW estimator can be viewed as a first-order bias correction of a plug-in estimator using the efficient influence function (Bickel et al., 1993; Tsiatis, 2006). The A-IPTW and other debiased (but asymptotically equivalent) estimators (van der Laan & Rubin, 2006) have recently been combined with various ML models (e.g., Shi et al., 2019; Hatt & Feuerriegel, 2021; Chernozhukov et al., 2022; Hines & Hines, 2025).

**Bayesian estimators.** Bayesian approaches to causal inference typically model the nuisance functions using flexible

---

[1]This bias is also referred to as regularization-induced confounding bias and is known to affect Bayesian non-parametric models (Hahn et al., 2018; Linero & Antonelli, 2023).

[2]Code is available at `https://github.com/Valentyn1997/freq-cons-pfns`.

nonparametric methods, such as Gaussian processes (Alaa & van der Schaar, 2017; 2018) and Bayesian regression trees (Chipman et al., 2010). For broader overviews, we refer to Li et al. (2023); Linero & Antonelli (2023). However, unlike frequentist estimators, Bayesian methods do *not* have a notion of (semi-parametric) efficiency. Instead, Bayesian estimators can have a related property, namely, frequentist consistency, which is formalized through the Bernstein–von Mises (BvM) theorem (Hartigan, 1983; Bickel & Kleijn, 2012).

**Bernstein–von Mises (BvM) theorem.** The BvM theorem states that Bayesian credible intervals for a causal estimand asymptotically coincide with confidence intervals centered at efficient frequentist estimators, such as the A-IPTW (Ray & van der Vaart, 2020). This property can be achieved in several ways. (a) One approach is to adopt tailored ad-hoc parametrizations of Bayesian models; for example, by placing priors directly on the conditional average treatment effect (Chipman et al., 2010), or by incorporating the propensity score as an input to the outcome model (Hahn et al., 2020). (b) A more general approach to obtaining the BvM property is by using one-step corrections based on efficient influence functions. Existing correction approaches for (b) include prior corrections (Ray & van der Vaart, 2020; Ray & Szabó, 2019), corrections to Bayesian variational objectives (Javurek et al., 2026), and posterior corrections (i.e., OSPC) (Yiu et al., 2025). We later employ posterior corrections, which can be implemented without re-training an amortized Bayesian model. Yet, to the best of our knowledge, the BvM property was not studied for PFNs (Müller et al., 2022), or, generally, for neural processes (Garnelo et al., 2018).

## 2.2. Prior-Data Fitted Networks (PFNs)

**PFNs.** PFNs were introduced as foundation models that amortize Bayesian inference by pretraining on synthetic datasets sampled from a user-specified prior over data-generating processes (Müller et al., 2022). A state-of-the-art example is TabPFN (Hollmann et al., 2023; 2025), which we use as our main backbone in our empirical analysis. In principle, any PFN such as TabPFN can be used for ATE estimation as an S- or T-learner (Künzel et al., 2019).

**PFNs for causal inference.** Recent work has extended the PFN paradigm to *direct* causal effect estimation by simulating causal structures during pretraining. Notable examples include Do-PFN (Robertson et al., 2025), CausalPFN (Balazadeh et al., 2025), and CausalFM (Ma et al., 2026). These works differ (i) in their underlying causal formulations (e. g., whether they adopt the potential outcomes framework); and (ii) in what posterior predictive densities (PPDs) and, consequently, what nuisance functions they are able to model. We provide a detailed comparison in Table 1. We later exclude Do-PFN from our analysis, as it overestimates uncertainty due to the fact that it relies on a *non-identifiable* formulation

(Ma et al., 2026).

**Uncertainty quantification with PFNs.** A key motivation for PFNs is that they can quantify an approximate total uncertainty of the outcome in the form of a PPD,[3] which can then be used for downstream tasks. However, deriving the uncertainty of the downstream estimands (such as conditional means or quantiles) is more challenging and requires an additional layer of inference (Jesson et al., 2025; Nagler & Rügamer, 2025). In particular, one must recover functional posteriors using the framework of *martingale posteriors (MPs)* (Fong et al., 2023). MPs can be constructed either solely based on PFNs (Ng et al., 2025) or by combining a PFN with an additional copula-based model (Nagler & Rügamer, 2025). In our work, we adopt the latter approach to recover the nuisance functions posteriors and, methodologically, extend the marginal point-wise MPs of Nagler & Rügamer (2025) to conditional MPs in order to obtain joint functional posteriors for the nuisance functions.

**Research gap.** To the best of our knowledge, no work has studied frequentist consistency of PFNs for causal inference or, more generally, semi-parametric estimation.

## 3. Preliminaries

**Notation.** We use uppercase letters $X, A, Y$ for random variables and lowercase $x, a, y$ for their realizations in $\mathcal{X}, \mathcal{A}, \mathcal{Y}$. For a generic random variable $Z$, $\mathbb{P}(Z)$ denotes its distribution; $\mathbb{E}(Z)$ is an expectation; and $P(z)$ refers to the corresponding density or probability mass (when needed, we also specify a random variable $Z$ in a lower index, i. e., $P_Z(z)$). Similarly, we define distributions in the functional space of the nuisance functions $\eta \in \mathcal{H}$. Namely, let $\Pi(\eta)$ denote a distribution over the nuisance functions space $\mathcal{H}$. Then, for a dataset $\mathcal{D} = \{z_i\}_{i=1}^n$, we denote the empirical distribution as $\mathbb{P}_n := n^{-1} \sum_{i=1}^n \delta_{z_i}$ where $\delta_z$ is a point mass distribution; and posterior distributions for random variables/functions as $\mathbb{P}(\cdot \mid \mathcal{D})/\Pi(\cdot \mid \mathcal{D})$, respectively. For a measurable function $f$, we write its $L_2$ norm as $\|f\| = (\mathbb{E}|f(Z)|^2)^{1/2}$, and we denote the empirical mean by $\mathbb{P}_n\{f(Z)\} := n^{-1} \sum_{i=1}^n f(z_i)$ and the mean wrt. Bayesian bootstrap process as $\mathrm{BB}_n\{f(Z)\} = \sum_{i=1}^n W_i f(z_i)$ where $(W_1, \ldots, W_n)$ is a sample from a Dirichlet process $\mathrm{Dir}(n; 1, \ldots, 1)$. The propensity score is $\pi(x) = \mathbb{P}(A = 1 \mid X = x)$, and we write the conditional outcome mean as $\mu_a(x) = \mathbb{E}(Y \mid X = x, A = a)$. When unambiguous, we use shorthand such as $\mathbb{E}(Y \mid x)$ and $\mathbb{P}(Y \mid x)$.

**Problem setup and causal estimand.** In our work, we rely on the potential outcomes framework (Rubin, 1974), where $Y[a]$ denotes the potential outcome under the intervention

---

[3]CausalPFN (Balazadeh et al., 2025) is a slight exception, as it provides not a PPD but a posterior density of the outcome model.

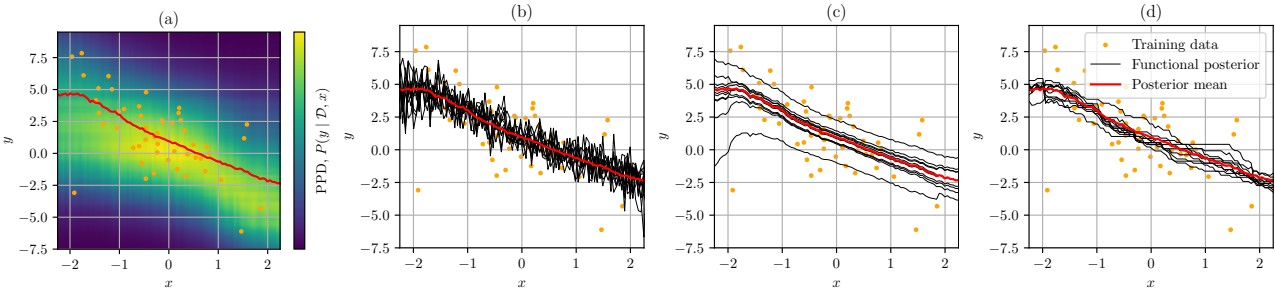

*Figure 1.* Recovering functional posteriors from TabPFN. In **(a)**, we show a PPD $P(y \mid \mathcal{D}, x)$, where $\mathcal{D} = \{(x_i, y_i)\}_{i=1}^{50}$. Then, in **(b)**-**(d)**, we draw PPD samples from different functional posteriors $\tilde{\mu} \sim \Pi(\mu \mid \mathcal{D})$ (recovered with martingale posteriors), where $\mu(x) = \mathbb{E}(Y \mid x)$. Notably, the same PPD $P(y \mid \mathcal{D}, x)$ **(a)** can encompass different functional posteriors (in the causal setting, those, in turn, lead to different ATE posteriors): **(b)** $x$-independent posterior, **(c)** $x$-parallel posterior, **(d)** smooth posterior.

$\mathrm{do}(A = a)$. We have access to the observational i.i.d. dataset $\mathcal{D} = \{(x_i, a_i, y_i)\}_{i=1}^{n} \sim \mathbb{P}(X, A, Y) = \mathbb{P}(Z)$. The covariates $X \in \mathcal{X} \subseteq \mathbb{R}^{d_x}$ are measured before treatment and may be high-dimensional, the treatment $A \in \{0, 1\}$ is binary, and the outcome $Y \in \mathcal{Y} \subseteq \mathbb{R}$ is real-valued. As a concrete example, in oncology studies, $Y$ could represent tumor size, $A$ whether radiotherapy was administered, and $X$ patient characteristics such as age and sex. Our target causal estimand is the average treatment effect (ATE) $\psi = \mathbb{E}(Y[1] - Y[0])$.

**Causal assumptions and identification.** To consistently estimate the ATE, we make standard causal assumptions (Rubin, 1974): (i) *consistency*: $Y[A] = Y$; (ii) *strong overlap*: $\mathbb{P}(\varepsilon < \pi(X) < 1 - \varepsilon) = 1$ for some $\varepsilon > 0$, and (iii) *unconfoundedness*: $(Y[0], Y[1]) \perp\!\!\!\perp A \mid X$. Then, under the assumptions (i)–(iii), the ATE is identified as

$$\psi = \psi(\eta) = \int_{\mathcal{X}} (\mu_1(x) - \mu_0(x)) \, \mathrm{d}P_X(x), \quad (1)$$

where $\eta = \{\mu_0, \mu_1, P_X\}$ are the ground-truth nuisance functions.

**Frequentist ATE estimation.** A naïve way to estimate the ATE is first (i) to fit the nuisance functions $\hat{\mu}_0, \hat{\mu}_1$ with some regression model (e.g., a neural network) and $\hat{P}_X = \mathbb{P}_n$ and then (ii) to plug them in into the identification formula from Eq. (1). This yields a *plug-in estimator*: $\psi^{\mathrm{PI}}(\hat{\eta}) = \mathbb{P}_n\{\hat{\mu}_1(X) - \hat{\mu}_0(X)\}$. Yet, this estimator suffers from a plug-in bias (Kennedy, 2024), which can be mitigated by the following bias-correction procedure:

$$\psi^{\mathrm{A\text{-}IPTW}}(\hat{\eta}) = \psi^{\mathrm{PI}}(\hat{\eta}) + \mathbb{P}_n\{\phi_\psi(Z; \hat{\eta})\}, \quad (2)$$

where $\psi^{\mathrm{A\text{-}IPTW}}$ is called an *augmented inverse probability of treatment weighted (A-IPTW) estimator*, and $\phi_\psi(Z; \hat{\eta})$ is an efficient influence function given by

$$\phi_\psi(Z; \eta = (\mu_0, \mu_1, \pi, P_X)) \quad (3)$$
$$= \frac{A - \pi(X)}{\pi(X)(1 - \pi(X))}(Y - \mu_A(X)) + \mu_1(X) - \mu_0(X) - \psi(\eta).$$

The core property of the A-IPTW estimator $\hat{\psi} = \psi^{\mathrm{A\text{-}IPTW}}(\hat{\eta})$ is that, under mild assumptions on the convergence of the nuisance functions (namely $R_2 =$

$\sqrt{n} \|\hat{\mu}_a - \mu_a\| \|\hat{\pi}^{-1} - \pi^{-1}\| \to 0$ for $a \in \{0, 1\}$) and proper sample splitting, it serves as a semi-parametric $\sqrt{n}$-consistent and efficient estimator of the ATE with the asymptotically normal distribution (Robins, 2000; Newey, 1994):

$$\sqrt{n}(\hat{\psi} - \psi) \xrightarrow{d} N(0; \mathrm{Var}[\phi_\psi(Z; \eta)]). \quad (4)$$

**Bayesian ATE estimation.** Bayesian semi-parametric estimators for ATE are conceptually similar to frequentist ones (Yiu et al., 2025) in that they also proceed in two stages. First, they assume some prior distribution for the nuisance functions $\eta \sim \Pi(\eta)$ (e.g., Gaussian processes for $\mu_a$ and an uninformative Dirichlet process prior for $P_X$). Then, they infer the posteriors $\tilde{\eta} \sim \Pi(\eta \mid \mathcal{D})$ (e.g., posterior Gaussian processes for $\mu_a$ and Bayesian bootstrap for $P_X$) and push-forward the latter through the mean functional from Eq. (1):

$$\psi^{\mathrm{PI}}(\tilde{\eta}) \mid \mathcal{D} = \mathrm{BB}_n\{\tilde{\mu}_1(X) - \tilde{\mu}_0(X)\}; \quad \tilde{\mu}_a \sim \Pi(\mu_a \mid \mathcal{D}), \ (5)$$

where posterior sampling from $\Pi(\mu_a \mid \mathcal{D})$ and Bayesian bootstrap are done independently (where we assume w.l.o.g. that the size of $\mathcal{D}$ and Bayesian bootstrap sample are both $n$). The sampling procedure from Eq. (5) then induces a posterior distribution of the ATE, $\mathbb{P}(\psi^{\mathrm{PI}}(\tilde{\eta}) \mid \mathcal{D})$, which we call a *plug-in posterior*.

**Frequentist consistency.** In the later sections, we are interested in the frequentist consistency of Bayesian ATE estimators. This property is desired, as it ensures asymptotically that (1) the result is indifferent to prior specification and (2) agreement between Bayesian and frequentist estimators. This can be formalized with the following property (Ray & van der Vaart, 2020; Yiu et al., 2025).

**Definition 1** (Semi-parametric Bernstein–von Mises (BvM) theorem). *A Bayesian estimator $\psi(\tilde{\eta}) \mid \mathcal{D}$ satisfies a semi-parametric BvM theorem if the posterior distribution of $\sqrt{n}(\psi(\tilde{\eta}) - \psi)$ converges in total variation (TV) to the asymptotic normal distribution of the A-IPTW estimator:*

$$d_{TV}\Big[\mathbb{P}(\sqrt{n}(\psi(\tilde{\eta}) - \psi) \mid \mathcal{D}); N(0; \mathrm{Var}[\phi_\psi(Z; \eta)])\Big] \to 0. \quad (6)$$

Notably, as we will demonstrate later, the plug-in posteriors generally do **not** satisfy the BvM theorem.

# 4. How to Turn PFNs into ATE Estimators?

To study the frequentist consistency of PFNs when used as *Bayesian* ATE estimators, we must first specify how PFNs can be turned into ATE estimators. Unlike classical semi-parametric Bayesian models, PFNs do not define explicit posteriors over nuisance functions. Instead, they provide only pointwise posterior predictive densities (PPDs), $P(\diamond \mid \mathcal{D}, x)$, for different components of the data-generating process (see Table 1 and Appendix A.1 for details). This seemingly minor distinction turns out to be crucial: depending on how PPDs are used, PFNs can behave either as (a) standard frequentist estimators or as (b) Bayesian estimators with fundamentally different uncertainty properties.

**(a) PFNs as frequentist estimators.** As a baseline, PFNs can be used in a purely frequentist manner by extracting point estimates from their posterior predictive densities. For example, given the conditional posterior mean estimator $\hat{\mu}_a(x) = \int_{\mathcal{Y}} y \, dP(y \mid \mathcal{D}, x, a)$ of TabPFN (Hollmann et al., 2023) or $\hat{\mu}_a(x) = \int_{\mathcal{Y}} \mu_a \, dP(\mu_a \mid \mathcal{D}, x)$ of CausalPFN (Balazadeh et al., 2025), either posterior mean estimator can be used to construct a plug-in ATE estimator. Similarly, we can further infer the posterior mean for the propensity score $\hat{\pi}(x) = P(A = 1 \mid \mathcal{D}, x)$ with TabPFN and use it inside the A-IPTW estimator (see Eq. (2)). Yet, PFNs are inherently Bayesian methods, and we thus prefer fully Bayesian estimators as shown in (b).

**(b) PFNs as Bayesian estimators.** A naïve way to use PFNs as Bayesian ATE estimators is to sample point-wise from the PPDs *as if they were functional posteriors*. For example, we might sample, independently for each $x$, the point-wise conditional outcome mean posterior from CausalPFN (Balazadeh et al., 2025) as $\tilde{\mu}_a(x) \sim \mathbb{P}(\mu_a \mid \mathcal{D}, x)$ and push it forward with Eq. (5). We call this ATE posterior an $x$-*independent PPD plug-in posterior*. Alternatively, we might sort the sampled $\tilde{\mu}_a(x)$ point-wise wrt. $\mathcal{Y}$ (e. g., ascending) and yield an $x$-*parallel PPD plug-in posterior*. Importantly, there is not a unique way to recover the nuisance function posteriors $\Pi(\eta \mid \mathcal{D})$ from the PPDs (see visual examples in Fig. 1). Still, both the independent and parallel posteriors misrepresent the uncertainty of the nuisance functions: the independent posterior fully discards a potential smooth structure of the nuisance functions, and the parallel posterior induces a non-existent dependency. Later, we will show that it is possible to recover smooth, "natural" posteriors of nuisance functions from PFNs using an MP framework.

Still, a more fundamental issue arises when PFNs are used with plug-in posteriors: *prior-induced confounding bias*, which we discuss next.

# 5. Frequentist Consistency of PFNs

In the following, we discuss ① what is a prior-induced confounding bias, ② how to correct it and achieve frequentist consistency, and ③ how to recover smooth posteriors when we want to employ PFNs as *Bayesian ATE estimators*. Towards the end, we introduce our main framework of martingale posteriors-based one-step posterior corrections (*MP-OSPC*).

## 5.1. ① Prior-Induced Confounding Bias

The plug-in ATE posterior, in general, faces a problem of a *prior-induced confounding bias (PICB)* (Hahn et al., 2018; Linero & Antonelli, 2023; Ray & van der Vaart, 2020). For example, when Bayesian non-parametric models (e. g., Gaussian processes (GPs)) are used to define the data-generating process for $\mathcal{D}$ (namely, $\mu_a$ and $\pi$), their prior rarely describes datasets with a high degree of *observed confounding* and, thus, *biases* the plug-in ATE posterior. The intuitive explanation of this phenomenon is that as the nuisance functions $\mu_a$ and $\pi$ are sampled from the prior, they rarely depend on the same subsets of covariates $x^J, J \subseteq \{1, \ldots, d_x\}$.

More generally, the PICB can break the consistency[4] of any Bayesian estimator as it induces either (i) an asymptotically *non-vanishing*, or (ii) *slower-than-$\sqrt{n}$ vanishing bias* in the ATE posterior. (i) The first situation can happen given a very general, non-smooth functional prior (e. g., Dirichlet), such that the likelihood does not dominate over the prior (Diaconis & Freedman, 1986). On the other hand, (ii) the second scenario is typical for smooth non-parametric models (e. g., GPs) (Ray & van der Vaart, 2020). Specifically, while the general consistency holds for these models, it is impossible for any regular Bayesian estimator to concentrate at a rate faster than $1/\sqrt{n}$ (van der Vaart, 2000).

Interestingly, existing PFNs are also affected by the PICB as they are trained on synthetic datasets sampled from a prior distribution. To show this, we sample the prior causal datasets, $\mathcal{D}_c = \{(x_i, a_i, y_i[0], y_i[1])\}_{i=1}^n$, and evaluate the degree of the observed confounding:

$$\Delta = \mathbb{E}(Y[1] - Y[0]) - \big(\mathbb{E}(Y \mid A = 1) - \mathbb{E}(Y \mid A = 0)\big), \quad (7)$$

which quantifies how the ATE differs from the difference in means (here, $Y$ is considered to be standard normalized). Fig. 2 illustrates the prior-induced confounding bias for different PFNs. We observed that, although the amount of the observed confounding $\Delta$ remains relatively fixed with increasing covariate dimensionality $d_x$ (unlike for GPs), $\Delta$ is still highly concentrated around zero for the PFNs.

Notably, the exact amount of the PICB in the ATE posterior

---

[4]Importantly, here we distinguish two types of consistency: (i) a *(general) consistency* and (ii) a *frequentist consistency*.

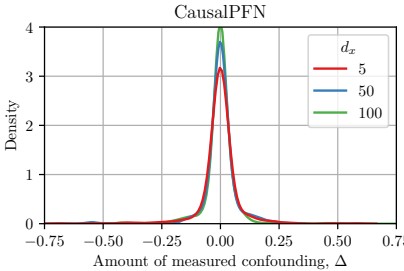
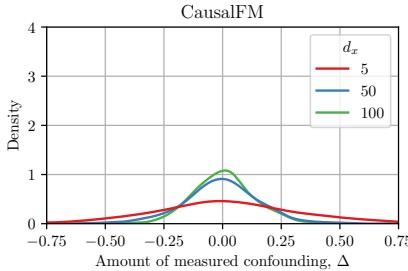

*Figure 2.* Prior-induced confounding bias of different PFNs. High values of $\Delta$ correspond to a high degree of the observed confounding. Thus, when $\Delta$ is concentrated around zero, the prior might induce the confounding bias which does not vanish asymptotically / vanishes very slowly with growing data. Here, we sample $B = 512$ causal datasets with $n = 10000$ each. $\Delta$ was computed from the prior causal datasets used to train PFNs: for CausalPFN/CausalFM, these are directly available in code or from the authors; for TabPFN, we additionally sampled treatment assignments in the same spirit as its synthetic classification data generation.

is intractable for PFNs (given that both the prior and the variational approximation are complex neural networks). Yet, while (i) the general consistency can be achieved by PFN-based plug-in estimators with sufficiently expressive priors (Balazadeh et al., 2025) so that the degree of observed confounding $\Delta$ has a wide enough support; (ii) the frequentist consistency still depends on the concentration of the prior distribution of $\Delta$.

*Takeaway:* The PICB is, therefore, a *main obstacle* for the frequentist consistency of the PFNs. That is, we cannot, in general, guarantee the BvM theorem as the prior has too much influence on the plug-in ATE posterior.

### 5.2. ② One-step Posterior Corrections

To circumvent the prior-induced confounding bias, we suggest employing a *one-step posterior correction (OSPC)* for the ATE plug-in posterior (Yiu et al., 2025) (this procedure is similar to the bias correction of the frequentist A-IPTW estimator). The OSPC defines the posterior for the ATE as the following push-forward of the posterior nuisance functions:

$$\psi^{\text{OSPC}}(\tilde{\eta}) \mid \mathcal{D} = \psi^{\text{PI}}(\tilde{\eta}) + \text{BB}_n\{\phi_\psi(Z; \tilde{\eta})\}; \tilde{\eta} \sim \Pi(\eta \mid \mathcal{D}), \quad (8)$$

where $\psi^{\text{PI}}(\tilde{\eta})$ is the plug-in ATE push-forward from Eq. (5), and $\phi_\psi$ is the efficient influence function of the ATE, see Eq. (3). The OSPC thus yields an *OSPC ATE posterior* $\mathbb{P}(\psi^{\text{OSPC}}(\tilde{\eta}) \mid \mathcal{D})$ given by the following push-forward:

$$\psi^{\text{OSPC}}(\tilde{\eta}) \mid \mathcal{D} = \text{BB}_n\left\{\xi_\psi(Z; \tilde{\eta})\right\}; \quad (9)$$

$$\xi_\psi(Z; \tilde{\eta}) = \frac{A - \tilde{\pi}(X)}{\tilde{\pi}(X)(1 - \tilde{\pi}(X))}(Y - \tilde{\mu}_A(X)) + \tilde{\mu}_1(X) - \tilde{\mu}_0(X);$$

$$\tilde{\eta} \sim \Pi(\eta \mid \mathcal{D}), \quad (10)$$

where $\xi_\psi$ is an uncentered efficient influence function. Notably, the Bayesian ATE estimation now requires modeling the functional posterior of the propensity score $\Pi(\pi \mid \mathcal{D})$, in addition to the outcome regressions. From a purely Bayesian perspective, this may appear unnecessary: the ground-truth causal estimand $\psi$ does *not* depend on the propensity score, and, thus, we do not need its posterior. We nevertheless incorporate the propensity score posterior for two fundamental

reasons: (i) to *resolve the formerly mentioned prior-induced confounding bias* as it increases uncertainty of the posterior in the regions of low overlap. Furthermore, (ii) by using the OSPC, we allow for the frequentist $\sqrt{n}$-consistency even when posteriors for both $\mu_a$ and $\pi$ concentrate at slower-than-parametric rates around the ground-truth, which yields a Bayesian analogue of *double-robustness*. Both (i) and (ii) are formalized in the following theorem.

**Theorem 1** (Semi-parametric BvM theorem of the OSPC ATE posterior)**.** *Assume the following holds for the observational data and a Bayesian estimator of the nuisance functions. Specifically, there exists a sequence of measurable subsets $H_n$ of $\mathcal{H}$ for which $\Pi(\tilde{\eta} \in H_n \mid \mathcal{D}) \to 1$ and for which (a)–(c) hold for $a \in \{0, 1\}$:*
*(a) $L_2$ concentration of $\tilde{\mu}_a$ and $\tilde{\pi}$:*

$$R_2 = \sqrt{n} \sup_{\tilde{\pi} \in H_n} \left\| \tilde{\pi}^{-1} - \pi^{-1} \right\| \cdot \sup_{\tilde{\mu}_a \in H_n} \|\tilde{\mu}_a - \mu_a\| \to 0. \quad (11)$$

*(b) Uniform bounding: for large $n$, there exists $C, \varepsilon > 0$ such that, for all $\tilde{\mu}_a, \tilde{\pi} \in H_n$, $\mathbb{P}(|Y - \tilde{\mu}_a(X)| < C) = 1$ and $\mathbb{P}(\varepsilon < \tilde{\pi}(X) < 1 - \varepsilon) = 1$.*
*(c) Donsker condition: nuisance functions posteriors are (informally) not too flexible (Yiu et al., 2025).*
*Then, the OSPC ATE posterior $\psi^{OSPC}(\tilde{\eta})$ from Eq (9) satisfies the BvM theorem (see Definition 1).*

*Proof.* This result is based on Theorem 4 of Yiu et al. (2025). We provide the sketch of the proof in Appendix A.3. □

Intuitively, the OSPC thus recovers the frequentist consistency of *any Bayesian ATE estimator of the nuisance functions* (as long as it satisfies the conditions of Theorem 1): it ensures that Bayesian and frequentist uncertainties are aligned in large samples so that prior specification does not influence the posterior asymptotically. Interestingly, if a Bayesian estimator is already debiased (e. g., CausalPFN/CausalFM may arguably perform debiasing out of the box as they were trained on the causal datasets), the OSPC does not change the plug-in posterior (see Appendix B.1 for further details).

In the following, we develop a framework so that the OSPC ATE posterior can also be implemented for the PFNs.

### 5.3. ③ Sampling Nuisance Functions with MPs

In the following, we aim to recover smooth functional posteriors $\tilde{\eta} \sim \Pi(\eta \mid \mathcal{D})$ from a PPD of some PFN, so that they can be used in the OSPC ATE posterior from Eq (9). For this, we use a predictive property of the PFNs together with an MP framework (Fong et al., 2023; Ng et al., 2025; Nagler & Rügamer, 2025). The MPs build on iteratively sampling and updating the PPDs:

Step 1 : $Z' \sim \mathbb{P}(Z \mid \mathcal{D}); \ \mathcal{D}' = \mathcal{D} \cup \{Z'\};$     (12)

Step 2 : $Z'' \sim \mathbb{P}(Z \mid \mathcal{D}'); \ \mathcal{D}'' = \mathcal{D}' \cup \{Z''\}; \dots$

Step N : $Z'^N \sim \mathbb{P}(Z \mid \mathcal{D}'^{N-1}); \ \mathcal{D}'^N = \mathcal{D}'^{N-1} \cup \{Z'^N\}; \dots$

and, then, using the random infinite dataset $\{Z', Z'', \dots\}$ equipped with a random density / probability mass function $P_\infty(z) \mid \mathcal{D}$ to infer posteriors of the downstream quantities. For example, we can infer the posterior for the mean $\theta(P_Z) = \int_{\mathcal{Z}} z \, dP_Z(z)$ via $\mathbb{P}(\theta \in A \mid \mathcal{D}) = \int_\Theta \mathbb{1}(\theta(P_Z) \in A) \, d\Pi(P_\infty(z) \mid \mathcal{D})$. We provide further background information on MPs in Appendix A.2.

**PFN-only MPs.** A direct way to recover the smooth functional posteriors for $\mu_a$ and $\pi$ is to employ PFNs as a posterior update model for MPs, as was done by the TabMGP (Ng et al., 2025). Concretely, for the posterior of $\mu_a$, we instantiate the generic MP updates at Step N from Eq. (12) using TabPFN as follows:

$$\begin{cases} (X, A)'^N \sim \mathbb{P}_n(X, A); \\ Y'^N \sim \mathbb{P}(Y \mid \mathcal{D}'^{N-1}, X'^N, A'^N); \end{cases} \quad (13)$$

where $\mathbb{P}_n(X, A)$ is the empirical distribution and $\mathbb{P}(Y \mid \mathcal{D}'^{N-1}, X'^N, A'^N)$ is given by the PPD of TabPFN. After $N$ updates, we draw one sample from $\Pi(\mu_a \mid \mathcal{D})$ by marginalizing the random PPD: $\tilde{\mu}_a = \int_{\mathcal{Y}} y \, dP(y \mid \mathcal{D}'^{N-1}, X'^N, A'^N)$. The functional posterior for the propensity score can be obtained from TabPFN analogously by iteratively updating $\mathbb{P}(A \mid \mathcal{D}'^{N-1}, X'^N)$. In principle, any other PFN can be used in place of TabPFN with this MP construction. However, none of the existing causal PFNs provide the PPDs for both the outcome and propensity score (see Table 1). Hence, we later use TabPFN as the backbone in our experiments.

However, the PFN-only MPs are problematic for two reasons. (1) PFNs often violate the so-called *martingale property* (i.e., the PPDs may add bias under sequential updating; see Nagler & Rügamer, 2025; Ng et al., 2025), and, thus, cannot be reliably used in the MP updates. (2) MP updates are time-consuming when large PFNs are used (e. g., transformer-based PFNs like TabPFN scale quadratically, that is, are in $O(N^2)$). To tackle both problems, we adopt a hybrid approach of combining PFNs and copulas (Nagler & Rügamer, 2025).

**PFN+copula MPs.** Nagler & Rügamer (2025) suggested combining the PFNs (used at Step 1) and copulas (used at Steps > 1), as originally suggested by Fong et al. (2023). Yet, they implemented the MP updates conditionally on $x$, while we want to recover the whole functional posteriors. Thus, we define the PFN+copula MP updates at Step $N$ as follows:

$$\begin{cases} V'^N = (X, A)'^N \sim \mathbb{P}_n(X, A); \\ u_N(v, V'^N) = 1 - \alpha_N(v, V'^N) + \alpha_N(v, V'^N) \, c_\rho(q_N, r_N); \\ P_N(y \mid \mathcal{D}'^N, x, a) = u_N(x, a, V'^N) \, P_{N-1}(y \mid \mathcal{D}'^{N-1}, x, a); \end{cases} \quad (14)$$

where $P_0(y \mid \mathcal{D}, x, a)$ is the PPD of TabPFN, $c_\rho(\cdot, \cdot)$ a bivariate copula density with correlation $\rho \in (0, 1)$, $\alpha_N$ is a learning rate, $q_N = F_{N-1}(y \mid \mathcal{D}'^{N-1}, x, a)$, and $r_N = F_N(Y'^N \mid \mathcal{D}'^{N-1}, V'^N)$ (where $F$ is a predictive posterior cumulative distribution function (CDF)). Then, the posterior of $\tilde{\mu}_a$ is obtained by the marginalization over $y$ (similarly to the PFN-only case). We provide further details on the specification of copulas and learning rates for $\mu_a$ and $\pi$ in Appendix D.

**Choice of functional posteriors.** Interestingly, the PFN+copula MPs combination allows us to recover the previously mentioned variants of functional posteriors from TabPFN (see Fig. 1). Namely, depending on whether we sample $r_N$ independently from $x$ and $y$, we can recover (a) $x$-independent posteriors (i.e., $r_N$ and $(x, y)$ are independent), (b) $x$-parallel posteriors (i.e., $r_N$ and $(x, y)$ are dependent), and (c) smooth posteriors (i.e., $r_N$ and only $x$ are dependent). We refer to Appendix D for further details on the three variants.

**Implementation details.** Combining MPs with the one-step posterior correction yields our main calibration procedure, which we denote by *MP-OSPC*. We illustrate *MP-OSPC* schematically in Fig. 6 in Appendix D. When instantiated with PFN+copula MPs, our *MP-OSPC* is flexible and computationally lightweight by introducing only a single hyperparameter $\rho \in (0, 1)$. Empirically, the performance of *MP-OSPC* is largely insensitive to the specific choice of $\rho$ and works equally well across several values for the ATE estimation. In all the experiments, we used $N = 100$ MP steps and $B = 100$ posterior draws. We refer to Appendix D for other implementation details.

### 5.4. BvM Property of MP-OSPC

To establish the BvM theorem for our *MP-OSPC*, the key requirement is to show that the $L_2$-concentration property (see Theorem 1) is satisfied for MP-based posteriors of a PFN. Intuitively, this condition ensures that the posterior concentrates sufficiently fast around the true nuisance functions so that the second-order remainder term vanishes asymp-

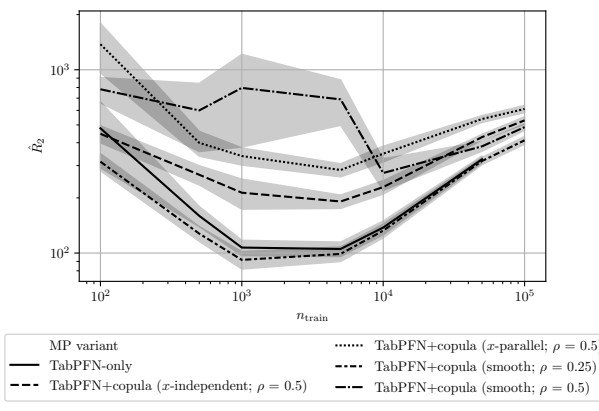

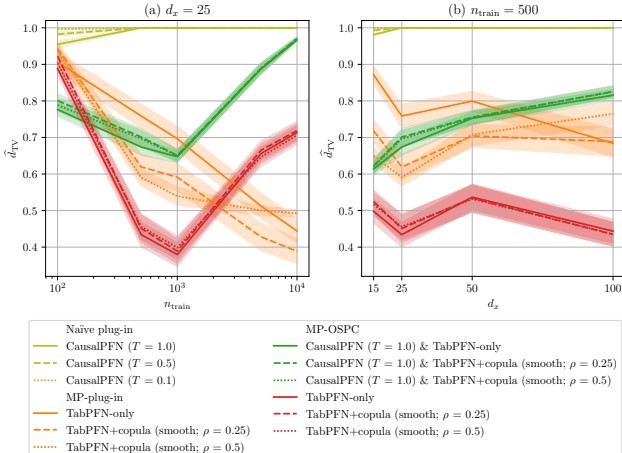

*Figure 3.* $L_2$-concentration check based on the synthetic data with varying size of the train data, $n_{\text{train}}$ (here: $d_x = 25$). Reported: mean $\hat{R}_2 \pm$ se over 10 runs (lower is better). Note that both x- and y-axes are log-scaled.

totically. Because PFNs are only approximately Bayesian, deriving the exact concentration rates for a specific PFN is challenging and depends on many factors (e. g., the underlying transformer architecture, the training procedure, and the specification of the synthetic priors).

Once the $L_2$-concentration condition is satisfied, the BvM theorem guarantees asymptotic normality. However, different constructions of functional posteriors (variants (a)–(c)) can still induce slightly different second-order contributions to the posterior variance. In particular, although all variants share the same leading variance term of order $1/n$, they differ by terms of order $o_{\mathbb{P}}(1/n)$. We provide further details in Appendix B.2.

A natural question is whether a failure of the BvM property should be attributed to the MP construction or to the underlying PFN. We view the dominant bottleneck as the latter (= PFN). Indeed, given an ideal PPD, MP uncertainty can be reduced by increasing the number of MP steps $N$; existing analyses of MP-based posterior sampling suggest a concentration speed depends on the learning rate $\alpha_N$ (Fong & Yiu, 2026; Nagler & Rügamer, 2025). In our case, $\alpha_N$ is chosen as $O(1/(n + N))$ (see Appendix D), and, hence, the MP-based posterior concentrates at the rate $O_{\mathbb{P}}(1/\sqrt{n}) + O_{\mathbb{P}}(1/\sqrt{N+n}) = O_{\mathbb{P}}(1/\sqrt{n})$ (Nagler & Rügamer, 2025). Hence, the second-order remainder $R_2$ is governed by how well the initial PFN-based PPD concentrates around the true density. Thus, in our setting, *violations of the BvM conditions should be seen as limitations of current PFNs rather than of the MP-OSPC wrapper.*

# 6. Experiments

**Setup.** We follow a standard causal benchmarking practice by using semi-synthetic datasets, where both counterfactual outcomes $Y[a]$ are available (and thus the ATE) (Curth & van der Schaar, 2021). Further, for fully-synthetic datasets, both nuisance functions $\mu_a$ and $\pi$ are known (and thus is the asymptotic variance of the A-IPTW estimator

*Figure 4.* Quality of the asymptotic uncertainty for Bayesian ATE estimators based on the synthetic data with (a) varying size of the train data, $n_{\text{train}}$, and (b) varying dimensionality of covariates, $d_x$. Reported: mean $\hat{d}_{\text{TV}} \pm$ se over 40 runs (lower is better).

from Eq. (4).[5] For all estimators, we perform a train-test data split, so that the nuisance functions are estimated with PFNs based on the train data, and the final ATE estimator is averaged/Bayesian-bootstrapped over the test data.

**Datasets.** In our experiments, we employ (i) the synthetic data generator of Curth & van der Schaar (2021) with varying $n$ and $d_x$ but with a fixed, large degree of the observed confounding $\Delta \approx -0.5$; (ii) IHDP dataset (Hill, 2011; Shalit et al., 2017) with $n = 672 + 75$, $d_x = 25$, and $\Delta \approx -0.02$; and (iii) 77 datasets from the ACIC 2016 datasets collection (Dorie et al., 2019) with $n = 4802$, $d_x = 82$, and varying $\Delta$. Further details are in Appendix E.

**Settings**. We performed the experiments in three different settings. (1) First, we tested how well different MP variants converge when one uses them to infer the functional posteriors (see Appendix F.1). (2) Then, in Sec. 6.1, we performed the check of the $L_2$-concentration property for PFN-based estimators. (3) Finally, in Sec. 6.2, we report the results for the ATE estimation.

## 6.1. $L_2$-Concentration Check

**Evaluation metric and MP variants.** Our aim is to empirically check whether the $L_2$-concentration condition holds for TabPFN when combined with the MPs, that is, whether the posterior concentrates sufficiently fast (in Appendix F.2, we additionally check the $L_2$-concentration of the CausalPFN & TabPFN combination). Hence, we evaluate an empirical analogue of the Bayesian second-order remainder $R_2$ from Eq. (11), i.e., $\hat{R}_2 = \sqrt{n} \max_{j=1,\ldots,B} \left\| \tilde{\pi}_j^{-1} - \pi^{-1} \right\| \cdot \sum_{a \in \{0,1\}} \max_{j=1,\ldots,B} \left\| \tilde{\mu}_{a,j} - \mu_a \right\|$, where $\tilde{\eta}_j$ is the $j$-th posterior draw obtained from a given MP variant. We consider several constructions of the MP posteriors: ***TabPFN-***

---

[5] For semi-synthetic datasets, we employ the propensity score estimated with TabPFN ($\hat{\pi}$ is a posterior mean).

***only*** (1 variant) and ***TabPFN+copula*** (4 variants with different choices of functional posteriors and $\rho$).

**Synthetic data results.** We show the results in Fig. 3. We find that the second-order remainder $\hat{R}_2$ is decreasing for nearly all the variants of the MP posteriors, as desired. Yet, when the dataset size is large (e. g., $n_{\text{train}} > 5000$), $\hat{R}_2$ starts to increase again. This appears to be a limitation of TabPFN in accurately recovering propensity score posteriors, particularly in regions where the true propensity scores are close to 0 or 1 (see the detailed concentration plots in Appendix F.2). As a consequence, TabPFN can only approximately satisfy the BvM theorem. In the subsequent experiments, we thus use *TabPFN+copula (smooth)* as a default variant of the MP, due to a better performance.

## 6.2. ATE Estimation

**Evaluation metrics.** We evaluate the uncertainty of the ATE estimators in two settings: (i) *asymptotic* and (ii) *finite-sample*. For (i), we estimate the empirical analogue of the total variation $\hat{d}_{\text{TV}}$ between a Bayesian estimator and the asymptotic distribution of the A-IPTW estimator from Eq. (4). For (ii), we assess the calibration of the credible intervals in the frequentist sense using a probability integral transform diagnostic: we repeat the experiment multiple times, evaluate the posterior ATE CDF at the ground-truth value, and then estimate the Kolmogorov-Smirnov distance $\hat{d}_{\text{KS}}$ between the random CDFs and a uniform distribution.

**Baselines.** In both settings (i) and (ii), we compare different Bayesian ATE estimators from three categories: (1) ***naïve plug-ins*** (namely, CausalPFN w/ different temperature $T$), (2) ***MP-plug-ins*** (= MP w/ the plug-in posterior), and (3) ***MP-OSPC*** (= MP w/ the OSPC posterior). Note that TabPFN and CausalFM cannot be used as (1) naïve plug-ins as they provide the total uncertainty and *not* the uncertainty of $\mu_a$. Also, in the latter two categories (2)–(3), we consider different backbones for modeling $\mu_a$ and $\pi$ with different hyperparameters (e. g., TabPFN or a combination of CausalPFN and TabPFN). We refer to Appendix A.1 for further details on the baselines.

**Synthetic data results.** The results for ATE estimation in (i) asymptotic setting are in Fig. 4. Therein, different variants of our *MP-OSPC* improve over plug-in estimators based on TabPFN and CausalPFN, and often achieve the best asymptotic alignment with the A-IPTW estimator for varying (a) dimensionality of covariates and (b) train data sizes. Notably, the slight drop in performance of *MP-OSPC* for large data sizes (e.g., $n_{\text{train}} > 1000$) is expected and can be attributed to a growing $R_2$ (see Fig. 3 and Fig. 8). Furthermore, we report the sensitivity to different hyperparameters ($\rho$, $B$, $N$) and (ii) finite-sample results in Appendix F.3.

**IHDP results.** The results for the IHDP dataset are in Appendix F.3, which support similar conclusions from above.

| $\mu_a$ model | $\pi$ model | ATE estimator | (i) %$_{\text{TV}}$ | (ii) %$_{\text{KS}}$ |
|---|---|---|---|---|
| CausalPFN ($T = 0.5$) | — | Naïve plug-in | **54.16** | 30.95 |
| | TabPFN+copula (smooth; $\rho = 0.25$) | MP-OSPC | 18.70 | **45.24** |
| | TabPFN+copula (smooth; $\rho = 0.5$) | MP-OSPC | 27.14 | 23.81 |
| CausalPFN ($T = 0.1$) | — | Naïve plug-in | **39.74** | 31.65 |
| | TabPFN+copula (smooth; $\rho = 0.25$) | MP-OSPC | 34.94 | **49.37** |
| | TabPFN+copula (smooth; $\rho = 0.5$) | MP-OSPC | 25.32 | 18.99 |
| TabPFN+copula (smooth; $\rho = 0.25$) | — | MP-plug-in | 14.81 | 20.45 |
| | TabPFN+copula (smooth; $\rho = 0.25$) | MP-OSPC | 33.25 | **34.09** |
| TabPFN+copula (smooth; $\rho = 0.5$) | — | MP-plug-in | 14.29 | 22.73 |
| | TabPFN+copula (smooth; $\rho = 0.5$) | MP-OSPC | **37.66** | 22.73 |

Higher = better (best in bold)

*Table 2.* Quality of (i) asymptotic and (ii) finite-sample uncertainties of Bayesian ATE estimators based on 77 ACIC-2016 datasets. Reported: (i) % of runs with the best performance wrt. $\hat{d}_{\text{TV}}$ over 77 datasets times 10 runs each (%$_{\text{TV}}$), and (ii) % of runs with the best performance wrt. $\hat{d}_{\text{KS}}$ over 77 datasets each evaluated with 10 runs (%$_{\text{KS}}$). Both percentages of (i) and (ii) are grouped wrt. the underlying PFN.

**ACIC 2016 results.** Table 2 shows pooled results across 77 datasets for both (i) and (ii) (we excluded TabPFN-only MP model from the experiments due to a too-long runtime). Overall, our *MP-OSPC* consistently improves upon the corresponding plug-in baselines in terms of both asymptotic alignment and finite-sample calibration. Notably, while our *MP-OSPC* does not substantially alter the results for CausalPFN, it consistently improves over TabPFN-based plug-in estimators. This can be attributed to (1) the distribution of confounding levels $\Delta$ in the ACIC 2016 datasets aligns well with the implicit PFN prior (cf. Fig. 2), in contrast to the synthetic setting; and (2) the fact that CausalPFN was trained with access to counterfactual data such that it oftentimes yields well-calibrated, already corrected posteriors for $\mu_a$ (we refer to Appendix F.3 for a more detailed overview of the ACIC 2016 results). Taken together, the results of the ACIC 2016 datasets confirm our theory that the OSPC does not degrade well-calibrated estimators and provides the greatest gains when the underlying PFN is miscalibrated.

## 7. Conclusion

**Limitations.** In our work, we discovered that current PFNs lack formal asymptotic guarantees (see, e. g., Sec. 6.1) and, thus, are mainly useful for *medium-sized datasets*. Yet, we argue that the key obstacle to BvM lies in the PFNs themselves rather than in our MP-OSPC wrapper.

**Extensions.** Here, we focus on the frequentist consistency of the ATE estimation with PFNs, and our work naturally extends to any *finite-dimensional target estimand*. On the other hand, the extension to the heterogeneous treatment effects is an open research problem: Namely, those are *infinitely-dimensional target estimands* and, hence, their frequentist uncertainty is not well-defined.

**Summary.** Our work is the first to provide a principled way to use PFNs as Bayesian ATE estimators so that they achieve frequentist consistency.

## Impact Statement

This paper presents work whose goal is to advance the field of machine learning. There are many potential societal consequences of our work, none of which we feel must be specifically highlighted here.

**Acknowledgments.** This paper is supported by the DAAD program "Konrad Zuse Schools of Excellence in Artificial Intelligence", sponsored by the Federal Ministry of Education and Research. S.F. acknowledges funding via Swiss National Science Foundation Grant 186932. This work has been supported by the German Federal Ministry of Education and Research (Grant: 01IS24082). RGK is supported by a Canada CIFAR AI Chair and a Canada Research Chair Tier II in Computational Medicine (CRC-2022-00049). This research was supported by an NFRF Special Call NFRFR2022-00526. Resources used in preparing this research were provided, in part, by the Province of Ontario, the Government of Canada through CIFAR, and companies sponsoring the Vector Institute.

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

# A. Background Materials

## A.1. Prior-data Fitted Networks

**PFNs as amortized Bayesian inference.** Prior-data fitted networks (PFNs) are foundation models that amortize Bayesian inference by pretraining on synthetic datasets sampled from a user-specified prior over data-generating processes (Müller et al., 2022). Concretely, let $\Pi_0$ denote the (synthetic) prior over data-generating processes $\mathcal{D} \sim \Pi_0(\mathcal{D})$, and let a sampled process induce a joint distribution for $(X, Y)$ (or, more generally, for $(X, A, Y)$ in our causal setting). For a dataset $\mathcal{D}$ and a query input $x$, Bayesian inference yields a posterior predictive density (PPD)

$$P(y \mid \mathcal{D}, x) = \int P(y \mid x, \theta) \, \mathrm{d}\Pi(\theta \mid \mathcal{D}), \tag{15}$$

where $\theta \in \Theta$ denotes latent parameters indexing the data-generating process (for the sake of generality, we assume that $\Theta$ can be an infinitely-dimensional (functional) space). A PFN is trained to approximate the mapping $(\mathcal{D}, x) \mapsto P(\cdot \mid \mathcal{D}, x)$ directly with a single forward pass, i.e., it learns a predictor that outputs a distribution $P_\theta(\cdot \mid \mathcal{D}, x) \approx P(\cdot \mid \mathcal{D}, x)$, where $\theta$ are parameters of the PFN. In practice, PFNs are implemented as conditional neural processes (Garnelo et al., 2018) (often transformer-based) and trained by minimizing the expected negative log-likelihood on synthetic tasks:

$$\theta^\star \in \arg\min_\theta \ \mathbb{E}_{\Pi_0} \mathbb{E}_{\mathcal{D}} \mathbb{E}_{(X,Y)} \big[ -\log P_\theta(Y \mid \mathcal{D}, X) \big]. \tag{16}$$

As a result, the PFN induces an *implicit prior* for any functional of interest, as it depends on the synthetic pretraining datasets distribution. When this implicit prior differs from the true data-generating process, it can systematically influence posterior predictions and downstream inference; in our setting, this distortion takes the form of prior-induced confounding bias studied in Sec. 5.1.

**Pointwise PPDs versus functional posteriors.** The defining output of a PFN is a *pointwise* PPD, $P(\diamond \mid \mathcal{D}, x)$, where $\diamond$ depends on what was used as the prediction target during pretraining. This is fundamentally different from classical Bayesian semi-parametric models, which define explicit functional posteriors $\Pi(\eta \mid \mathcal{D})$ over nuisance functions $\eta$. Given a PPD, we can always extract posterior means pointwise; for instance, for a regression-type PPD, we have

$$\hat{\mu}(x) = \int_{\mathcal{Y}} y \, \mathrm{d}P(y \mid \mathcal{D}, x), \tag{17}$$

and, for a binary target, we obtain posterior mean probabilities analogously. However, the OSPC posterior from Sec. 5.2 requires sampling *entire nuisance functions* $\tilde{\eta} \sim \Pi(\eta \mid \mathcal{D})$. Importantly, PFNs do *not* provide $\Pi(\eta \mid \mathcal{D})$ directly, and there is *not* a unique way to reconstruct a functional posterior from the same collection of pointwise PPDs (see Fig. 1 for illustrative examples). This motivates our use of martingale posteriors in Appendix A.2.

**PFNs for causal inference and the outputs in Table 1.** Recent work trains PFNs directly for causal estimands by simulating causal structures during pretraining, including Do-PFN (Robertson et al., 2025), CausalPFN (Balazadeh et al., 2025), and CausalFM (Ma et al., 2026). These methods differ in what *posterior distribution* they output in the sense of Table 1:

- **TabPFN.** TabPFN (Hollmann et al., 2023; 2025) is a general-purpose PFN for tabular prediction. When used as a nuisance learner in causal inference settings (marked by [†] in Table 1), we treat $(X, A)$ as inputs and $Y$ as the label, so the PFN provides an outcome PPD

$$P(y \mid \mathcal{D}, x, a), \tag{18}$$

  which we interpret as a pointwise predictive distribution for $Y[a]$ given $(x, a)$. Its posterior mean yields $\hat{\mu}_a(x) = \int_{\mathcal{Y}} y \, \mathrm{d}P(y \mid \mathcal{D}, x, a)$ and the plug-in CATE mean $\hat{\tau}(x) = \hat{\mu}_1(x) - \hat{\mu}_0(x)$. In addition, by running TabPFN in classification mode with label $A$ and features $X$, we obtain a pointwise treatment PPD, $P(A = a \mid \mathcal{D}, x)$, and hence $\hat{\pi}(x) = P(A = 1 \mid \mathcal{D}, x)$.

- **Do-PFN.** Do-PFN (Robertson et al., 2025) provides a PPD for counterfactual outcomes $Y[a]$ (Table 1) but does *not* provide a propensity-score PPD but relies on a formulation that can be *non-identifiable* (marked by [‡]). Following Ma et al. (2026), we thus exclude Do-PFN from our analysis.

- **CausalPFN.** CausalPFN (Balazadeh et al., 2025) departs from label-level PPDs and instead outputs a posterior distribution over conditional outcome mean, in particular, a pointwise density $P(\mu_a \mid \mathcal{D}, x)$ (and thus also for $\tau$ through

$\mu_1 - \mu_0$). Because it does *not* provide a label-level PPD $P(Y[a] \mid \mathcal{D}, x)$ (nor a propensity-score PPD), it *cannot* be used with the MP constructions studied here (Table 1).

- **CausalFM.** CausalFM (Ma et al., 2026) outputs a pointwise PPD for the treatment effect itself, i. e., the PPD of a distribution $\mathbb{P}(\Delta = Y[1] - Y[0] \mid \mathcal{D}, x)$, namely $P(\delta \mid \mathcal{D}, x)$. While this is useful for effect prediction, it does *not* provide the joint set of nuisance-function posteriors needed for our OSPC-based ATE inference. Furthermore, it also *cannot* be combined with the MPs, as the outputs $Y[1] - Y[0]$ are not the part of the input dataset $\mathcal{D}$ (thus, a pointwise posterior density of CATE, $P(\tau \mid \mathcal{D}, x)$, cannot be recovered). Therefore, CausalFM *cannot* even be used as a naïve Bayesian plug-in estimator of the ATE.

**Choice of baselines.** Table 1 summarizes which PPDs and posterior means are available from each PFN "out of the box." In our work, TabPFN and CausalPFN are the only two baselines that can correctly provide the uncertainty of the outcome model, $\mu_a$ – and, thus, the Bayesian uncertainty of the ATE. Yet, TabPFN is the only model that provides PPDs for both the outcome and the propensity models and is therefore viable for the OSPC; we then recover *functional* posteriors from these pointwise PPDs using martingale posteriors (Appendix A.2). For the overview of connections between different PFNs and the estimators they yield, we refer to Fig. 5.

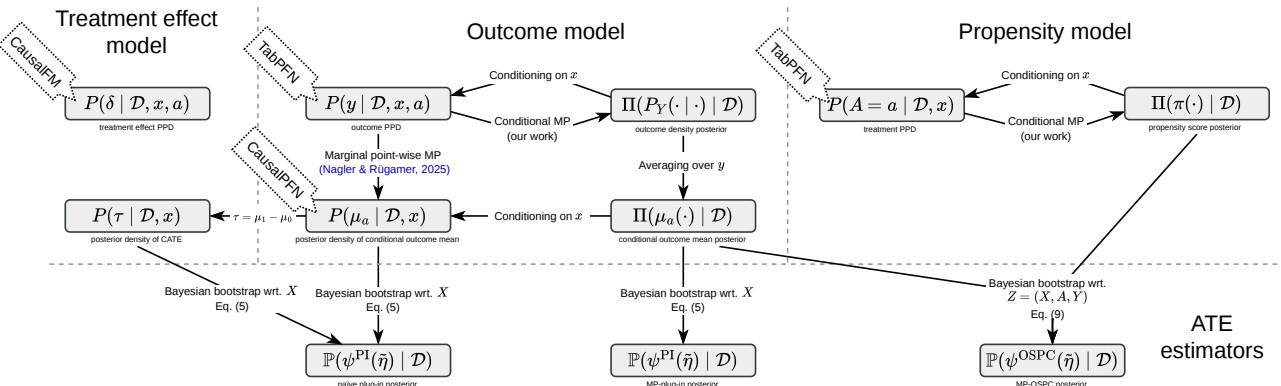

*Figure 5.* Overview of connections between different posterior distributions, PPDs, and the underlying PFNs.

## A.2. Martingale Posteriors

Martingale posteriors (MPs) provide a general way to construct posterior distributions for parameters and functionals using *only* a posterior predictive density (PPD) that can be sequentially updated, but without requiring an explicit likelihood or prior (Fong et al., 2023). In our setting, this is useful because PFNs naturally provide PPDs, while our OSPC construction requires draws from posteriors over nuisance functions.

**Setup.** Let $Z$ be a generic random variable with distribution $\mathbb{P}(Z)$ and density / probability mass function $P(z)$. We observe an i.i.d. dataset $\mathcal{D} = \{z_i\}_{i=1}^n$. Assume we have access to a PPD $P(z \mid \mathcal{D})$ that can be updated after augmenting the dataset with additional observations.

**MP updates and the random limit law.** MPs iteratively sample pseudo-observations from the current PPD and update the dataset, as in Eq. (12). Concretely, define $\mathcal{D}'^0 = \mathcal{D}$ and, for $N \geq 1$,

$$Z'^N \sim \mathbb{P}(Z \mid \mathcal{D}'^{N-1}), \qquad \mathcal{D}'^N = \mathcal{D}'^{N-1} \cup \{Z'^N\}. \tag{19}$$

Let $\mathcal{F}_N = \sigma(\mathcal{D}'^N)$ be the induced filtration and define the sequence of predictive laws $P_N(\cdot) = \mathbb{P}(Z \in \cdot \mid \mathcal{D}'^N)$. A key condition is the *martingale property*: for every measurable set $B \subseteq \mathcal{Z}$,

$$\mathbb{E}[P_{N+1}(B) \mid \mathcal{F}_N] = P_N(B). \tag{20}$$

When Eq. (20) holds, $\{P_N(B)\}_{N \geq 0}$ is a bounded martingale and therefore converges almost surely with $N \to \infty$ to a random limit $P_\infty(B)$. Collecting these limits over measurable $B$ yields a random probability measure $P_\infty(\cdot) \mid \mathcal{D}$, which can be viewed as an (implicit) posterior draw over the data-generating distribution.

**Induced posteriors for functionals.** MPs induce posteriors for any downstream functional $\theta(P_Z) \in \Theta$ by pushforward through the random limit law $P_\infty(\cdot) \mid \mathcal{D}$. For example, for $\theta(P_Z) = \int_{\mathcal{Z}} z \, dP_Z(z)$, the MP posterior satisfies

$$\mathbb{P}(\theta \in A \mid \mathcal{D}) = \int \mathbb{1}(\theta(P_\infty) \in A) \, d\Pi(P_\infty \mid \mathcal{D}), \tag{21}$$

where $\Pi(P_\infty \mid \mathcal{D})$ denotes the distribution of the random measure $P_\infty(\cdot) \mid \mathcal{D}$ induced by the MP updates.

**Connection to Bayes posteriors.** If the PPD $P(z \mid \mathcal{D})$ is itself the posterior predictive density of some Bayesian model with prior $\Pi(P_Z)$, then the MP construction recovers the corresponding Bayes posterior over $P_Z$ in the sense that $P_\infty(\cdot) \mid \mathcal{D}$ has the same distribution as a draw from $\Pi(P_Z \mid \mathcal{D})$ (Fong et al., 2023). More generally, MPs remain well-defined whenever the sequential updates satisfy the martingale property, even if the underlying likelihood/prior are not available in closed form.

**Nuisance-function posteriors as MP functionals.** In the potential outcomes setup, the nuisance functions are functionals of the joint law $P_Z$:

$$\pi(x) = \mathbb{P}(A = 1 \mid X = x), \qquad \mu_a(x) = \mathbb{E}(Y \mid X = x, A = a).$$

Thus, a draw $P_\infty(\cdot) \mid \mathcal{D}$ immediately induces draws $\tilde{\pi}$ and $\tilde{\mu}_a$ by evaluating these functionals at $P_\infty$. In practice, we approximate $\Pi(P_\infty \mid \mathcal{D})$ by running Eq. (19) for a finite number of MP steps $N$ and repeating the procedure to obtain multiple posterior draws, as used by our *MP-OSPC* implementation (cf. Fig. 6 in Appendix D). Finally, when the base predictor violates Eq. (20) (as may happen for PFNs under repeated updating), sequential MP updates can accumulate bias (Ng et al., 2025; Nagler & Rügamer, 2025), motivating the hybrid MP constructions we employ in the main text.

### A.3. Semi-parametric Bernstein–von Mises Theorem

In the following, we briefly sketch the statement and the proof of the semi-parametric Bernstein–von Mises theorem from Yiu et al. (2025).

**Theorem 1** (Semi-parametric BvM theorem of the OSPC ATE posterior). *Assume the following holds for the observational data and a Bayesian estimator of the nuisance functions. Specifically, there exists a sequence of measurable subsets $H_n$ of $\mathcal{H}$ for which $\Pi(\tilde{\eta} \in H_n \mid \mathcal{D}) \to 1$ and for which (a)–(c) hold for $a \in \{0, 1\}$:*
*(a) $L_2$ concentration of $\tilde{\mu}_a$ and $\tilde{\pi}$:*

$$R_2 = \sqrt{n} \sup_{\tilde{\pi} \in H_n} \left\| \frac{1}{\tilde{\pi}} - \frac{1}{\pi} \right\| \cdot \sup_{\tilde{\mu}_a \in H_n} \|\tilde{\mu}_a - \mu_a\| \to 0. \tag{22}$$

*(b) Uniform bounding: for large $n$, there exists $C, \varepsilon > 0$ such that, for all $\tilde{\mu}_a, \tilde{\pi} \in H_n$, $\mathbb{P}(|Y - \tilde{\mu}_a(X)| < C) = 1$ and $\mathbb{P}(\varepsilon < \tilde{\pi}(X) < 1 - \varepsilon) = 1$.*
*(c) Donsker condition:[6] a Donsker condition holds for $H_n$ (i. e., nuisance functions posteriors are not too flexible) (Yiu et al., 2025).*
*Then, the OSPC ATE posterior $\psi^{OSPC}(\tilde{\eta})$ from Eq (9) satisfies the BvM theorem, namely:*

$$d_{TV}\left[ \mathbb{P}(\sqrt{n}(\psi^{OSPC}(\tilde{\eta}) - \psi) \mid \mathcal{D}); N(0; \text{Var}[\phi_\psi(Z; \eta)]) \right] \to 0. \tag{23}$$

*Proof (sketch).* Let $\hat{\psi} = \psi + \mathbb{P}_n\{\phi_\psi(Z; \eta)\}$ denote the A-IPTW estimator of the ATE. Consider the high-posterior-probability set $H_n$ from the theorem statement, with $\Pi(\tilde{\eta} \in H_n \mid \mathcal{D}) \to 1$. With $H_n$, it is enough to prove the result conditional on $\tilde{\eta} \in H_n$.

Using the OSPC posterior,

$$\psi^{OSPC}(\tilde{\eta}) = \psi^{PI}(\tilde{\eta}) + \text{BB}_n\{\phi_\psi(Z; \tilde{\eta})\}, \tag{24}$$

we make the following decomposition:

$$\sqrt{n}\big(\psi^{OSPC}(\tilde{\eta}) - \hat{\psi}\big) = \sqrt{n}(\text{BB}_n - \mathbb{P}_n)\{\phi_\psi(Z; \tilde{\eta})\} + \mathbb{G}_n[\phi_\psi(Z; \tilde{\eta}) - \phi_\psi(Z; \eta)] - \sqrt{n}R_2(\eta, \tilde{\eta}) \tag{25}$$

where $\mathbb{G}_n = \sqrt{n}(\mathbb{P}_n - \mathbb{E})$, and $R_2(\eta, \tilde{\eta})$ is the second-order remainder.

---

[6](c) Donsker condition can also be replaced by sample splitting, given additional sup-convergence and envelope conditions (see Assumption 1(c)* in Yiu et al. (2025)).

Now, we can upper-bound/control the three terms.

First, for $\tilde{\eta}$, the Bayesian bootstrap central limit theorem yields:

$$\sqrt{n}(\mathrm{BB}_n - \mathbb{P}_n)\{\phi_\psi(Z; \tilde{\eta})\} \rightsquigarrow N(0, \mathrm{Var}[\phi_\psi(Z; \tilde{\eta})]). \tag{26}$$

Second, by the Donsker condition (Assumption (c)),

$$\sup_{\tilde{\eta} \in H_n} |\mathbb{G}_n[\phi_\psi(Z; \tilde{\eta}) - \phi_\psi(Z; \eta)]| = o_P(1). \tag{27}$$

Third, for the ATE the remainder satisfies $|R_2(\eta, \tilde{\eta})| \lesssim \max_{a \in \{0,1\}} ||\tilde{\mu}_a - \mu_a|| \cdot ||\tilde{\pi}^{-1} - \pi^{-1}||$, hence by Assumption (a) implies

$$\sqrt{n} \sup_{\tilde{\eta} \in H_n} |R_2(\eta, \tilde{\eta})| \to 0. \tag{28}$$

Finally, Assumption (a) also yields

$$\sup_{\tilde{\eta} \in H_n} ||\phi_\psi(\cdot; \tilde{\eta}) - \phi_\psi(\cdot; \eta)|| \to 0, \tag{29}$$

so that $\mathrm{Var}[\phi_\psi(Z; \tilde{\eta})] \to \mathrm{Var}[\phi_\psi(Z; \eta)]$.

Therefore, $\sqrt{n}(\psi^{\mathrm{OSPC}}(\tilde{\eta}) - \hat{\psi}) \mid \mathcal{D} \rightsquigarrow N(0, \mathrm{Var}[\phi_\psi(Z; \eta)])$, which proves the semi-parametric BvM theorem. $\qquad\square$

# B. Questions & Answers

## B.1. What if the OSPC Is Applied to an Already Calibrated Model?

If a posterior $\tilde{\mu}_a$ is already calibrated (in our frequentist consistency sense), applying the OSPC essentially has no asymptotic effect, since $\mathrm{BB}_n\{\phi_\psi(Z;\tilde{\eta})\} \approx 0$. For example, CausalPFN may yield approximately debiased posteriors $\tilde{\mu}_a$ for datasets with not too large measured confounding $\Delta \in (-0.25, 0.25)$. The reason for this is that the debiasing term can be upper-bounded by the amount of measured confounding. To see that, we denote $\bar{\mu}_a := \mathbb{E}(Y \mid A = a)$, $\mu_A := A\bar{\mu}_1 + (1-A)\bar{\mu}_0 = \mathbb{E}(Y \mid A)$, and $w_\pi(Z) := \frac{A-\pi(X)}{\pi(X)(1-\pi(X))}$, and show the following:

$$|\mathbb{E}\{\phi_\psi(Z;\tilde{\eta})\}| = |\mathbb{E}\left[w_\pi(Z)\{Y - \tilde{\mu}_A(X)\} + \tilde{\mu}_1(X) - \tilde{\mu}_0(X) - \psi(\tilde{\eta})\right]| = |\mathbb{E}\left[w_\pi(Z)\{Y - \tilde{\mu}_A(X)\}\right]| \tag{30}$$

$$\leq |\mathbb{E}\left[w_\pi(Z)\{Y - \mu_A\}\right]| + \underbrace{|\mathbb{E}\left[w_\pi(Z)\{\mu_A - \tilde{\mu}_A(X)\}\right]|}_{\text{(negligible if } \tilde{\mu}_a \text{ is well calibrated)}} \tag{31}$$

$$\lesssim \left|\mathbb{E}\left[\frac{A-\pi(X)}{\pi(X)(1-\pi(X))}\{Y - \mu_A\}\right]\right| = \left|\mathbb{E}\left[\frac{A}{\pi(X)}\{Y - \bar{\mu}_1\}\right] - \mathbb{E}\left[\frac{1-A}{1-\pi(X)}\{Y - \bar{\mu}_0\}\right]\right| \tag{32}$$

$$= \left|\mathbb{E}\left[\frac{\mathbb{E}\{A(Y - \bar{\mu}_1) \mid X\}}{\pi(X)}\right] - \mathbb{E}\left[\frac{\mathbb{E}\{(1-A)(Y - \bar{\mu}_0) \mid X\}}{1-\pi(X)}\right]\right| \tag{33}$$

$$= |\mathbb{E}\left[\mu_1(X) - \bar{\mu}_1\right] - \mathbb{E}\left[\mu_0(X) - \bar{\mu}_0\right]| \tag{34}$$

$$= |\mathbb{E}\{\mu_1(X) - \mu_0(X)\} - \{\mathbb{E}(Y \mid A = 1) - \mathbb{E}(Y \mid A = 0)\}| \tag{35}$$

$$= |\mathbb{E}\{Y[1] - Y[0]\} - \{\mathbb{E}(Y \mid A = 1) - \mathbb{E}(Y \mid A = 0)\}| = |\Delta|. \tag{36}$$

where $\mu_A = \mathbb{E}(Y \mid A)$ is a treatment-conditional mean, and $\tilde{\eta}$ is assumed to contain the ground-truth propensity score $\pi$. Hence, the correction term $\mathrm{BB}_n\{\phi_\psi(Z;\tilde{\eta})\}$ is close to zero, so the OSPC does not alter the resulting estimator.

## B.2. How to Choose a Functional Posterior?

In the following, we formalize the distinction between different variants (a)-(c) of functional posteriors wrt. their asymptotic performance.

**Proposition 1** (Asymptotic variance of MP-based posteriors). *Assume an MP-based Bayesian estimator $\tilde{\eta}^\diamond$ (where $\diamond$ is either a (a) $x$-independent, (b) $x$-parallel, or (c) smooth variant of the MP-based posterior) satisfies the conditions of Theorem 1. Then, the asymptotic variance of the MP-based OSPC ATE posterior is approximately*

$$\mathrm{Var}[\psi^{OSPC}(\tilde{\eta}^\diamond) \mid \mathcal{D}] \approx \frac{\mathrm{Var}[\phi_\psi(Z;\eta)]}{n} + o_\mathbb{P}(1/n) + \delta_n^\diamond, \tag{37}$$

*where*
- $\delta_n^{(a)} = O_\mathbb{P}(i_n) = O_\mathbb{P}\left(\mathbb{P}_n\{V(Z)\}/n\right)$ *and* $V(Z) = \mathrm{Var}_{\tilde{\eta}}[\xi_\psi(Z;\tilde{\eta}^{(a)}) \mid \mathcal{D}]$;
- $\delta_n^{(b)} = O_\mathbb{P}(p_n) = O_\mathbb{P}\left(\mathbb{P}_n^2\{C^{(b)}(Z,Z')\}\right) \leq O_\mathbb{P}\left(\mathbb{P}_n\{V(Z)\}\right)$ *and* $C^{(b)}(Z,Z') = \mathrm{Cov}_{\tilde{\eta}}[\xi_\psi(Z;\tilde{\eta}^{(b)}), \xi_\psi(Z';\tilde{\eta}^{(b)}) \mid \mathcal{D}] > 0$;
- $\delta_n^{(c)} = O_\mathbb{P}(s_n), 0 \leq s_n \leq p_n$.

*Furthermore, $\delta_n^\diamond = o_\mathbb{P}(1/n)$, under the assumption (a) of Theorem 1 (i.e., $L_2$-concentration).*

*Proof.* We refer to Appendix C. $\square$

Proposition 1 implies that three variants induce OSPC ATE posteriors with different contributions to the asymptotic variance. Under (a) $x$-independent coupling, the contribution of the nuisance uncertainty to the ATE uncertainty is the lowest and decreases with the Bayesian bootstrap (BB) sample size at a rate $1/n$. Conversely, under (b) $x$-parallel coupling, this contribution stays constant wrt. the BB sample size and can only be reduced by contracting posteriors with increasing size of $\mathcal{D}$. Finally, (c) smooth coupling also provides a constant, yet smaller than (b) contribution to the ATE uncertainty, which also decreases with growing $\mathcal{D}$. Arguably, the (c) smooth MP posterior would be the *most "natural" choice* to represent the posterior uncertainty of nuisance functions: (1) it preserves the original smoothness of the underlying PFN, unlike (a) $x$-independent posteriors; and (2) it does not introduce additional dependency wrt. $x$ (and thus does not additionally inflate the variance for ATE posterior), unlike (b) $x$-parallel posteriors.

## C. Theoretical Results

**Proposition 1** (Asymptotic variance of MP-based posteriors)**.** *Assume an MP-based Bayesian estimator $\tilde{\eta}^\diamond$ (where $\diamond$ is either a (a) $x$-independent, (b) $x$-parallel, or (c) smooth variant of the MP-based posterior) satisfies the conditions of Theorem 1. Then, the asymptotic variance of the MP-based OSPC ATE posterior is approximately*

$$\text{Var}[\psi^{OSPC}(\tilde{\eta}^\diamond) \mid \mathcal{D}] \approx \frac{\text{Var}[\phi_\psi(Z; \eta)]}{n} + o_\mathbb{P}\left(\frac{1}{n}\right) + \delta_n^\diamond, \tag{38}$$

*where*

- $\delta_n^{(a)} = O_\mathbb{P}(i_n) = O_\mathbb{P}\big(\mathbb{P}_n\{V(Z)\}/n\big)$ *and* $V(Z) = \text{Var}_{\tilde{\eta}}[\xi_\psi(Z; \tilde{\eta}^{(a)}) \mid \mathcal{D}]$;
- $\delta_n^{(b)} = O_\mathbb{P}(p_n) = O_\mathbb{P}\big(\mathbb{P}_n^2\{C^{(b)}(Z, Z')\}\big) \le O_\mathbb{P}\big(\mathbb{P}_n\{V(Z)\}\big)$ *and* $C^{(b)}(Z, Z') = \text{Cov}_{\tilde{\eta}}[\xi_\psi(Z; \tilde{\eta}^{(b)}), \xi_\psi(Z'; \tilde{\eta}^{(b)}) \mid \mathcal{D}] > 0$;
- $\delta_n^{(c)} = O_\mathbb{P}\big(s_n\big), 0 \le s_n \le p_n$.

*Furthermore, $\delta_n^\diamond = o_\mathbb{P}(1/n)$, under the assumption (a) of Theorem 1 (i.e., $L_2$-concentration).*

*Proof.* By definition in Eq. (9), the OSPC posterior is

$$\psi^{\text{OSPC}}(\tilde{\eta}^\diamond) \mid \mathcal{D} = \text{BB}_n\left\{\xi_\psi(Z; \tilde{\eta}^\diamond)\right\} = \sum_{i=1}^n W_i\, \xi_\psi(Z_i; \tilde{\eta}^\diamond) = \mathbf{W}^T \mathbf{\Xi}^\diamond, \tag{39}$$

where $\tilde{\eta}^\diamond \sim \Pi(\eta \mid \mathcal{D})$ is a functional nuisance posterior, $\mathbf{W} = (W_1, \ldots, W_n)^T \sim \text{Dir}(n; 1, \ldots, 1)$ independently of $\tilde{\eta}^\diamond$ and $Z$, and $\mathbf{\Xi}^\diamond = (\xi_\psi(Z_1; \tilde{\eta}^\diamond), \ldots, \xi_\psi(Z_n; \tilde{\eta}^\diamond))^T$.

Then, by the law of total variance:

$$\text{Var}[\psi^{\text{OSPC}}(\tilde{\eta}^\diamond) \mid \mathcal{D}] = \mathbb{E}_{\mathbf{W}}\Big[\text{Var}_{\tilde{\eta}}\big[\mathbf{W}^T\mathbf{\Xi}^\diamond \mid \mathcal{D}, \mathbf{W}\big] \mid \mathcal{D}\Big] + \text{Var}_{\mathbf{W}}\Big[\mathbb{E}_{\tilde{\eta}}\big[\mathbf{W}^T\mathbf{\Xi}^\diamond \mid \mathcal{D}, \mathbf{W}\big] \mid \mathcal{D}\Big]. \tag{40}$$

Because $\mathbf{W}$ is independent of $\mathbf{\Xi}^\diamond$ given $\mathcal{D}$:

- $\mathbb{E}_{\tilde{\eta}}\big[\mathbf{W}^T\mathbf{\Xi}^\diamond \mid \mathcal{D}, \mathbf{W}\big] = \mathbf{W}^T\mathbf{m}$, where $\mathbf{m}_i = M(Z_i) := \mathbb{E}_{\tilde{\eta}}[\xi_\psi(Z_i; \tilde{\eta}^\diamond) \mid \mathcal{D}]$, and $M(Z_i)$ does not depend on how the posterior $\tilde{\eta}$ is coupled and, thus, on a variant $\diamond$;

- $\text{Var}_{\tilde{\eta}}\big[\mathbf{W}^T\mathbf{\Xi}^\diamond \mid \mathcal{D}, \mathbf{W}\big] = \mathbf{W}^T\mathbf{C}^\diamond\mathbf{W}$, where $\mathbf{C}_{ij}^\diamond = C^\diamond(Z_i, Z_j) = \text{Cov}_{\tilde{\eta}}[\xi_\psi(Z_i; \tilde{\eta}^\diamond), \xi_\psi(Z_j; \tilde{\eta}^\diamond) \mid \mathcal{D}]$, $\mathbf{C}_{ii}^\diamond = V(Z_i) = \text{Var}_{\tilde{\eta}}[\xi_\psi(Z_i; \tilde{\eta}^\diamond) \mid \mathcal{D}]$, and $V(Z_i)$ does not depend on how the posterior $\tilde{\eta}$ is coupled and, thus, on a variant $\diamond$.

Therefore, the following holds

$$\text{Var}[\psi^{\text{OSPC}}(\tilde{\eta}^\diamond) \mid \mathcal{D}] = \underbrace{\mathbb{E}_{\mathbf{W}}\big[\mathbf{W}^T\mathbf{C}^\diamond\mathbf{W} \mid \mathcal{D}\big]}_{\text{(i) BB-weight variability}} + \underbrace{\text{Var}_{\mathbf{W}}\big[\mathbf{W}^T\mathbf{m} \mid \mathcal{D}\big]}_{\text{(ii) nuisance-posterior variability}}. \tag{41}$$

Now, under the assumptions of Theorem 1, we can easily show that the second term (ii) matches the variance of the efficient influence function. That is, by properties of the Dirichlet distribution:

$$\text{Var}_{\mathbf{W}}\big[\mathbf{W}^T\mathbf{m} \mid \mathcal{D}\big] = \frac{n\sum_{i=1}^n \mathbf{m}_i^2 - (\sum_{i=1}^n \mathbf{m}_i)^2}{n^2(n+1)} = \frac{1}{n+1}\,\mathbb{P}_n\big\{(M(Z) - \mathbb{P}_n\{M(Z)\})^2\big\} \tag{42}$$

$$\approx \frac{1}{n}\,\text{Var}[\phi_\psi(Z; \eta)] + o_\mathbb{P}\left(\frac{1}{n}\right). \tag{43}$$

However, the term (i) differs for different variants $\diamond$ (here we use the properties of the Dirichlet distribution again):

$$\delta_n^\diamond = \mathbb{E}_{\mathbf{W}}\big[\mathbf{W}^T\mathbf{C}^\diamond\mathbf{W} \mid \mathcal{D}\big] = \frac{2}{n(n+1)}\sum_{i=1}^n V(Z_i) + \frac{1}{n(n+1)}\sum_{i,j=1; i\neq j}^n C^\diamond(Z_i, Z_j). \tag{44}$$

For $\diamond =$ (a) $x$-independent coupling, all the off-diagonal covariance terms are zero (i. e., $C^{(a)}(Z_i, Z_j) = 0$), as the values $\xi_\psi(Z_i; \tilde\eta^{(a)})$ and $\xi_\psi(Z_j; \tilde\eta^{(a)})$ are independent for i.i.d. $Z_i, Z_j$. Hence, for $\diamond =$(a) $x$-independent coupling, the term (i) is

$$\delta_n^{(a)} = \mathbb{E}_\mathbf{W}\big[\mathbf{W}^T\mathbf{C}^{(a)}\mathbf{W} \mid \mathcal{D}\big] = \frac{2}{n+1}\mathbb{P}_n\{V(Z)\} = O_\mathbb{P}(\mathbb{P}_n\{V(Z)\}/n). \tag{45}$$

For $\diamond =$ (b) $x$-parallel coupling, the following holds:

$$\delta_n^{(b)} = \mathbb{E}_\mathbf{W}\big[\mathbf{W}^T\mathbf{C}^{(b)}\mathbf{W} \mid \mathcal{D}\big] = \frac{1}{n+1}\mathbb{P}_n\{V(Z)\} + \frac{n}{n+1}\mathbb{P}_n^2\{C^{(b)}(Z, Z')\} = O_\mathbb{P}(\mathbb{P}_n^2\{C^{(b)}(Z, Z')), \tag{46}$$

and, given our construction of $x$-parallel coupling (= co-monotonicity of $\xi_\psi(Z; \tilde\eta^{(b)})$ wrt. $\tilde\eta$), $C^{(b)}(Z, Z') > 0$ for any $Z, Z' \in \mathcal{Z}$. Furthermore, by Cauchy-Schwarz, we can upper-bound the last term of Eq. (46) as

$$\mathbb{P}_n^2\{C^{(b)}(Z, Z')\} \le \frac{1}{n^2}n\sum_{i=1}^n V(Z_i) = \mathbb{P}_n\{V(Z)\}. \tag{47}$$

Eq. (46) also applies to $\diamond =$ (c) smooth coupling:

$$\delta_n^{(c)} = O_\mathbb{P}(\mathbb{P}_n^2\{C^{(c)}(Z, Z')). \tag{48}$$

Here, however, the terms $C^{(c)}(Z, Z') > 0$ for $Z$ and $Z'$ spatially close to each other, and $C^{(c)}(Z, Z') \approx 0$ for far away points. Therefore, the following inequality holds:

$$0 \le \mathbb{P}_n^2\{C^{(c)}(Z, Z')\} \le \mathbb{P}_n^2\{C^{(b)}(Z, Z')\}. \tag{49}$$

Finally, we show that $\delta_n^\diamond = o_\mathbb{P}(1/n)$ under the assumption (a) of Theorem 1 (i.e., $L_2$-concentration). As shown previously, we have

$$\delta_n^\diamond = \frac{2}{n(n+1)}\sum_{i=1}^n V(Z_i) + \frac{1}{n(n+1)}\sum_{i,j=1;i\ne j}^n C^\diamond(Z_i, Z_j) \tag{50}$$

$$= \frac{1}{n(n+1)}\sum_{i=1}^n V(Z_i) + \frac{1}{n(n+1)}\sum_{i,j=1}^n C^\diamond(Z_i, Z_j) \tag{51}$$

$$= \frac{1}{n+1}\mathbb{P}_n\{V(Z)\} + \frac{1}{n(n+1)}\operatorname{Var}_{\tilde\eta}\Big[\sum_{i=1}^n \xi_\psi(Z_i; \tilde\eta^\diamond) \mid \mathcal{D}\Big] \tag{52}$$

$$= \frac{1}{n+1}\mathbb{P}_n\{V(Z)\} + \frac{n}{n+1}\operatorname{Var}_{\tilde\eta}[\mathbb{P}_n\{\xi_\psi(Z_i; \tilde\eta^\diamond)\} \mid \mathcal{D}]. \tag{53}$$

Here, the first term, $\frac{1}{n+1}\mathbb{P}_n\{V(Z)\} = o_\mathbb{P}(1/n)$, given the uniform bounding assumption of Theorem 1. The latter term can be upper-bounded by

$$\operatorname{Var}_{\tilde\eta}[\mathbb{P}_n\{\xi_\psi(Z_i; \tilde\eta^\diamond)\} \mid \mathcal{D}] = \operatorname{Var}_{\tilde\eta}[\mathbb{P}_n\{\xi_\psi(Z_i; \tilde\eta^\diamond) - \xi_\psi(Z_i; \eta)\} \mid \mathcal{D}] \tag{54}$$

$$\le \mathbb{E}_{\tilde\eta}\big[\mathbb{1}\{\tilde\eta^\diamond \in H_n\} \cdot \mathbb{P}_n\{\xi_\psi(Z; \tilde\eta^\diamond) - \xi_\psi(Z; \eta)\}^2 \mid \mathcal{D}\big] + \mathbb{E}_{\tilde\eta}\big[\mathbb{1}\{\tilde\eta^\diamond \notin H_n\} \cdot \mathbb{P}_n\{\xi_\psi(Z; \tilde\eta^\diamond) - \xi_\psi(Z; \eta)\}^2 \mid \mathcal{D}\big], \tag{55}$$

Then, by using a standard second-order remainder expansion + empirical-process term for ATE for $\tilde\eta^\diamond \in H_n$, we yield

$$|\mathbb{P}_n\{\xi_\psi(Z; \tilde\eta^\diamond) - \xi_\psi(Z; \eta)\}| \lesssim \max_{a\in\{0,1\}} \sup_{\tilde\pi^\diamond \in H_n}\left\|\frac{1}{\tilde\pi^\diamond} - \frac{1}{\pi}\right\| \cdot \sup_{\tilde\mu_a^\diamond \in H_n}\|\tilde\mu_a^\diamond - \mu_a\| = o_\mathbb{P}(1/\sqrt{n}), \tag{56}$$

and, by using the law of iterated expectations, we can further update the inequality in Eq. (54):

$$\operatorname{Var}_{\tilde\eta}[\mathbb{P}_n\{\xi_\psi(Z_i; \tilde\eta^\diamond)\} \mid \mathcal{D}] = \Pi(\tilde\eta^\diamond \in H_n) \cdot o_\mathbb{P}(1/n) + O_\mathbb{P}(\Pi(\tilde\eta^\diamond \notin H_n)). \tag{57}$$

Finally, we can choose such concentration sets $H_n$ such that $\Pi(\tilde\eta^\diamond \notin H_n) = o_\mathbb{P}(1/n)$ (e. g., a contraction ball with exponentially small posterior tails), so the following holds:

$$\operatorname{Var}_{\tilde\eta}[\mathbb{P}_n\{\xi_\psi(Z_i; \tilde\eta^\diamond)\} \mid \mathcal{D}] = o_\mathbb{P}(1/n). \tag{58}$$

$\square$

# D. Implementation Details

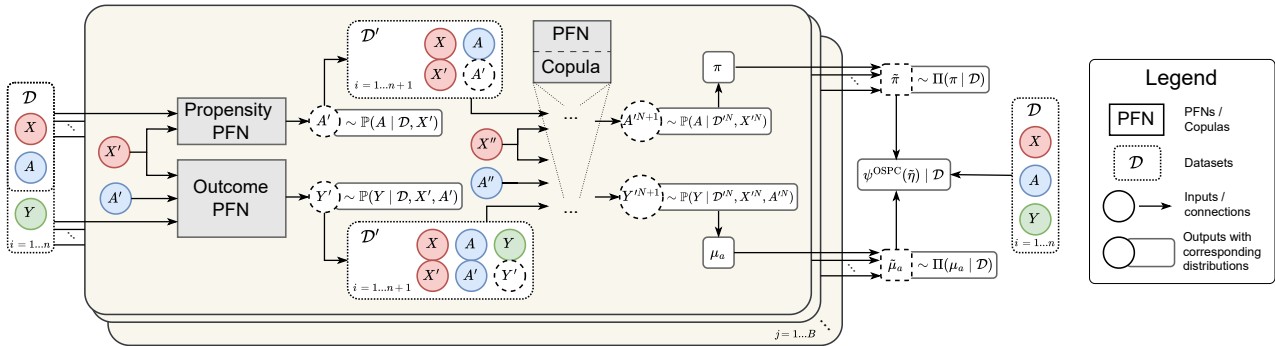

*Figure 6.* Overview of our *MP-OSPC* calibration procedure. Here, $N$ is the number of MP steps and $B$ is the number of posterior draws from the functional posteriors. Our *MP-OSPC* with PFNs thus yields the OSPC ATE posterior and serves as a Bayesian ATE estimator.

## D.1. Copula-based MPs Details

This section provides additional definitions for Eq. (14), especially for the Gaussian copula density $c_\rho$ and the weighting sequence $\alpha_N$ used in the copula-based MP updates (Fong et al., 2023).

**Gaussian copula density $c_\rho$ (conditional outcome mean).** In Eq. (14), $c_\rho(u, v)$ denotes the *bivariate Gaussian copula density* with correlation parameter $\rho \in (0, 1)$. Writing $\Phi$ and $\phi$ for the standard normal CDF and density, and $\phi_2(\cdot, \cdot; \rho)$ for the standard bivariate normal density with correlation $\rho$, the Gaussian copula density is

$$c_\rho(u, v) = \frac{\phi_2\big(\Phi^{-1}(u),\ \Phi^{-1}(v); \rho\big)}{\phi\big(\Phi^{-1}(u)\big)\, \phi\big(\Phi^{-1}(v)\big)}, \qquad (u, v) \in (0, 1)^2. \tag{59}$$

Intuitively, $c_\rho$ interpolates between an independent update (when $\rho \to 0$) and a sharply data-adaptive kernel (when $\rho \to 1$), with larger $\rho$ producing less smoothing (Fong et al., 2023).

**Beta–Bernoulli mixture copula (propensity score).** For the propensity model, we used the following beta–Bernoulli mixture copula (instead of $c_\rho(u, v)$):

$$d_\rho(u, v) = \begin{cases} 1 - \rho + \rho \frac{\min\{u, v\}}{uv}, & \text{if } a = A'^N, \\ 1 - \rho + \rho \frac{u - \min\{u, 1-v\}}{uv}, & \text{if } a \neq A'^N. \end{cases} \tag{60}$$

**The update weight $\alpha_N$.** The scalar $\alpha_N \in (0, 1)$ in Eq. (14) controls the contribution of the copula term relative to the previous predictive, and it is also the primary driver of posterior uncertainty under predictive resampling (Fong et al., 2023). A sufficient condition for the resulting predictive sequence to stabilize is $\alpha_N = O((n + N)^{-1})$. Following Fong et al. (2023), we use the sequence

$$\alpha_N(v, V'^N) = \frac{\alpha_N \prod_{j=1}^{d_v} c_\rho(\Phi(v_j), \Phi(V_j'^N))}{1 - \alpha_N + \alpha_N \prod_{j=1}^{d_v} c_\rho(\Phi(v_j), \Phi(V_j'^N))}, \qquad \alpha_N = \left(2 - \frac{1}{n + N}\right) \frac{1}{n + N + 1}, \qquad N \geq 1, \quad (61)$$

which decays as $O((n + N)^{-1})$ but more slowly than the commonly used $\alpha_N = (n + N + 1)^{-1}$. Empirically, this choice yields substantially better uncertainty calibration under predictive resampling (Fong et al., 2023).

**Replacing $r_N$ by a uniform draw (conditional outcome mean).** In Eq. (14), the term $r_N$ is a *probability integral transform (PIT)* quantity of the form $r_N = F_N(Y'^N \mid \mathcal{D}'^{N-1}, V'^N)$. In the continuous outcome case, if $Y'^N \sim \mathbb{P}(Y \mid \mathcal{D}'^{N-1}, V'^N)$, then

$$r_N = F_N(Y'^N \mid \mathcal{D}'^{N-1}, V'^N) \sim \text{Unif}(0, 1). \tag{62}$$

Therefore, for the outcome posterior, we may equivalently draw $r_N \sim \text{Unif}(0, 1)$, avoiding an explicit draw of $Y'^N$ and any subsequent evaluation of the predictive CDF (cf. Eq. (4.7) in Fong et al. (2023)).

**Coupling pointwise posteriors into a functional posterior.** Different choices of how we sample $r_N$ lead to different *functional* posteriors while yielding the same pointwise marginal $P_N(y \mid \mathcal{D}'^N, x, a)$:

- *(a) $x$-independent posteriors.* For evaluation points $(x_1, \ldots, x_M)$ and $(y_1, \ldots, y_K)$, draw $r_{N,1,1}, \ldots, r_{N,M,K} \overset{\text{i.i.d.}}{\sim}$ Unif$(0,1)$ independently of both $(x_1, \ldots, x_M)$ and $(y_1, \ldots, y_K)$.

- *(b) $x$-parallel posteriors.* For evaluation points $(x_1, \ldots, x_M)$ and $(y_1, \ldots, y_K)$, draw a single $r_N \overset{\text{i.i.d.}}{\sim}$ Unif$(0,1)$.

- *(c) smooth posteriors.* For evaluation points $(x_1, \ldots, x_M)$ and $(y_1, \ldots, y_K)$, draw $r_{N,1}, \ldots, r_{N,K} \overset{\text{i.i.d.}}{\sim}$ Unif$(0,1)$ independently of $(y_1, \ldots, y_K)$.

All three constructions share the same pointwise posteriors $P_N(y \mid \mathcal{D}'^N, x, a)$; they differ only in how uncertainty is coupled across $x$ and $y$.

### D.2. Other Implementation Details

We implement our *MP-OSPC* in NumPy and PyTorch. To stabilize the ATE estimation, we truncate too low propensity scores $\pi(x) < 0.05$ in both the ground-truth asymptotic variance of the A-IPTW estimator from Eq. (4) and the OSPC ATE posterior from Eq. (9).

# E. Dataset Details

## E.1. Synthetic Dataset

We use the simulation design from Curth & van der Schaar (2021) (Appendix D.1), where we focus on setting (ii), which combines confounding with a non-trivial, nonlinear treatment effect. Covariates are generated as $X \in \mathbb{R}^{d_x}$ with independent standard normal components and are partitioned into disjoint subsets, including confounders $X_C$ and outcome-only covariates $X_O$ (each of dimension 5), as well as treatment-effect covariates $X_\tau$ (dimension 5). The conditional average potential outcome (CAPO) of the control is defined as a quadratic form, and treatment is assigned according to a nonlinear propensity score. In setting (ii), the treated CAPO adds a quadratic treatment-effect component,

$$\mu_1(x) = \mu_0(x) + \mathbf{1}^\top x_\tau^2,$$

so that $\tau(x) = \mu_1(x) - \mu_0(x) = \mathbf{1}^\top x_\tau^2$, and observed outcomes are generated as $Y = A\mu_1(X) + (1 - A)\mu_0(X) + \varepsilon$ with $\varepsilon \sim \mathcal{N}(0, 1)$.

To study higher-dimensional regimes beyond the original design (which uses $d_x = 25$), we augment the covariates by appending additional independent noise features, sampled via $X^{\text{aug}} \sim \mathcal{N}(0, 1)$, and define the final covariate vector as $X = [X_{\text{orig}}, X^{\text{aug}}]$. These added covariates do not enter $\pi(x)$, $\mu_0(x)$, or $\mu_1(x)$, and therefore increase $d_x$ without changing the underlying treatment assignment mechanism or outcome-generating mechanism.

## E.2. IHDP Dataset

The Infant Health and Development Program (IHDP) benchmark (Hill, 2011; Shalit et al., 2017) is a widely used semi-synthetic dataset for evaluating treatment effect estimation methods. It provides 100 fixed train–test splits with $n_{\text{train}} = 672$ training units and $n_{\text{test}} = 75$ test units, each described by $d_x = 25$ covariates. In the data-generating mechanism, the ground-truth CAPOs take the form of an exponential outcome model under control, $\mu_0(x)$, and a linear outcome model under treatment, $\mu_1(x)$. A known limitation of IHDP is its pronounced lack of overlap, which can make propensity-score-based methods (e. g., the A-IPTW estimator) numerically unstable (Curth & van der Schaar, 2021; Curth et al., 2021).

## E.3. ACIC 2016 Datasets

The ACIC 2016 benchmark (Dorie et al., 2019) constructs covariates from the large-scale Collaborative Perinatal Project on developmental disorders (Light, 1973). The resulting datasets vary along several dimensions, including (i) the number of true confounders, (ii) the degree of overlap, and (iii) the smoothness and functional form of the ground-truth CAPOs. Overall, ACIC 2016 comprises 77 distinct data-generating processes, and for each process it provides 100 samples of equal size. After one-hot encoding categorical variables, each sample contains $n = 4802$ units with $d_x = 82$ covariates.

# F. Additional Experiments

## F.1. MP Convergence

As shown in Nagler & Rügamer (2025), the martingale property does not hold for TabPFN but does hold for copulas. Still, since the martingale property is sufficient but not necessary for MP convergence (Ng et al., 2025), we conducted empirical diagnostics. Specifically, in Fig. 7, we demonstrate the convergence plots for different variants of functional MPs depending on the number of MP steps $N$. Here, we see that, with few exceptions for large $\rho$, the posterior standard deviation generally flattens with the growing number of MP steps (e. g., for $N = 300$ with $n = 100$, for $N = 250$ with $n = 500$, etc.). Importantly, even though it takes up to $N = 300$ for the full MP convergence (in some cases even more), our MP-OSPC allows for good ATE estimation already for $N = 100$ (as we later demonstrate in one of the experiments in Appendix F.3).

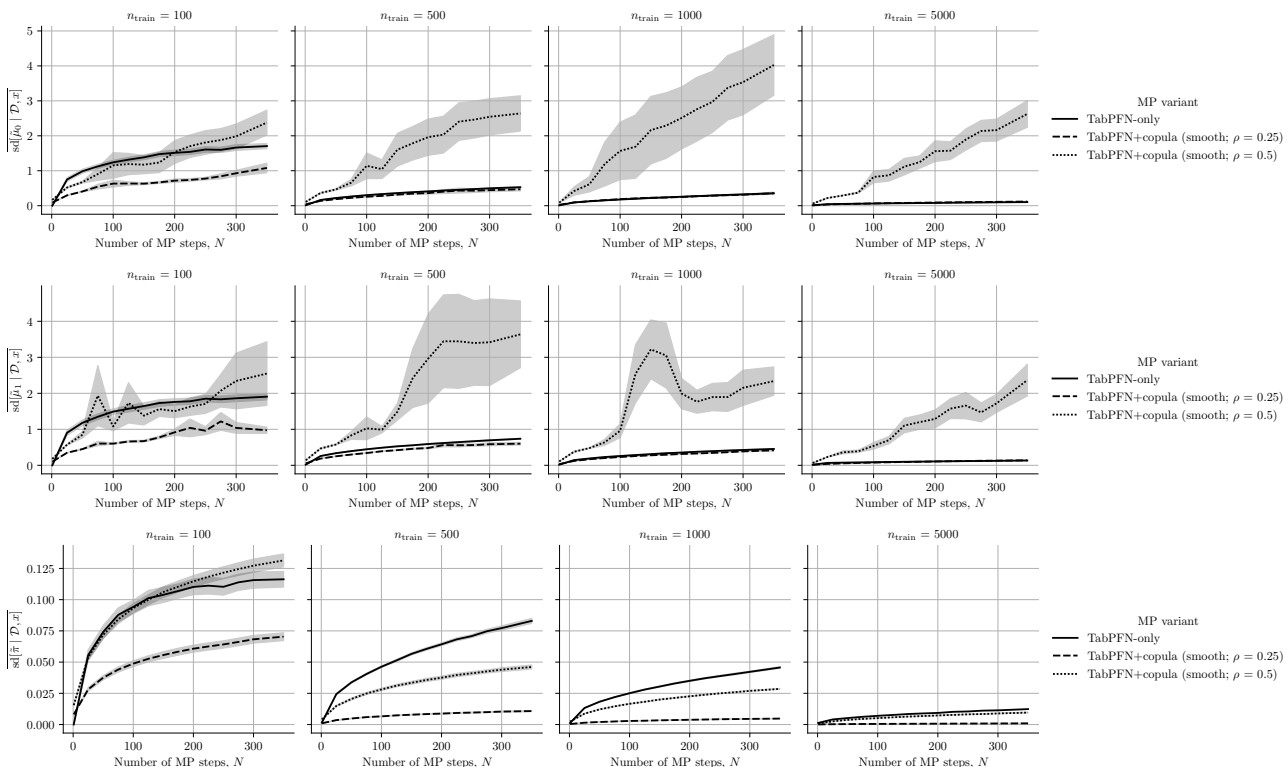

*Figure 7.* Convergence of different MP variants wrt. increasing number of MP steps $N$ for every nuisance function based on the synthetic data with varying size of the train data, $n_{\text{train}}$ (here $d_x = 25$). Reported: mean posterior standard deviation $\overline{\text{sd}[\diamond \mid \mathcal{D}, x]} \pm$ se over 10 runs, where $\diamond$ corresponds to different nuisance functions: (top row) $\diamond = \mu_0$, (middle row) $\diamond = \mu_1$, and (bottom row) $\diamond = \pi$.

## F.2. $L_2$-Concentration Check

### F.2.1. DETAILED $L_2$-CONCENTRATION CHECK (TABPFN)

In Fig. 8, we demonstrate the posterior concentration rates ($L_2$-concentration) of TabPFN with different MP variants for every nuisance function based on the synthetic data. Here, the nuisance specific errors $\hat{E}$ (that contribute to the second order remainder $\hat{R}_2$) reach their lowest value when $n_{\text{train}} = 5000$ for conditional outcome mean posteriors ($\tilde{\mu}_a$), or already when $n_{\text{train}} = 1000$ for inverse propensity score posterior ($\tilde{\pi}^{-1}$). These results indicate that performance improves with sample size up to a moderate regime, consistent with the $L_2$-concentration condition required for the BvM theorem. *However, beyond this regime, the concentration of the inverse propensity score posterior deteriorates, which reflects a difficulty of accurately recovering inverse propensity scores for propensities close to zero and one with TabPFN.* Together, this confirms that, in practice, the contraction rates are sufficient to yield stable and well-aligned uncertainty estimates, yet only for medium-sized datasets.

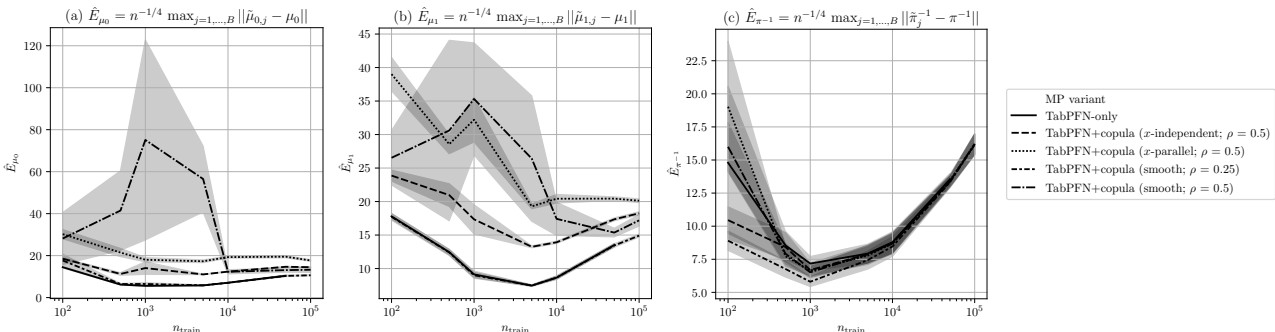

*Figure 8.* $L_2$-concentration check for every nuisance function based on the synthetic data with varying size of the train data, $n_{\text{train}}$ (here $d_x = 25$). Reported: mean $L_2$ error $\hat{E}_\diamond \pm$ se over 10 runs (lower is better), where $\diamond$ corresponds to different nuisance functions: (a) $\diamond = \mu_0$, (b) $\diamond = \mu_1$, and (c) $\diamond = \pi^{-1}$.

### F.2.2. Detailed $L_2$-Concentration Check (CausalPFN & TabPFN)

Fig. 9 shows the detailed $L_2$-concentration results for the combination of CausalPFN and TabPFN (where CausalPFN is used as outcome models, and TabPFN is used as a propensity model). These results indicate that performance degrades with increasing sample size for CausalPFN-based functional posteriors $\mu_a$, as seen in (a)-(b). This is not surprising, considering that CausalPFN was trained on synthetic datasets with up to $n \approx 2000$. However, as seen in (c), the second-order remainder $\hat{R}_2$ converges for $n < 1000$. This hints at the double robustness property: the divergence of the CausalPFN-based posteriors for $\mu_a$ can be partially compensated with a concentration of TabPFN-based propensity score MP posterior.

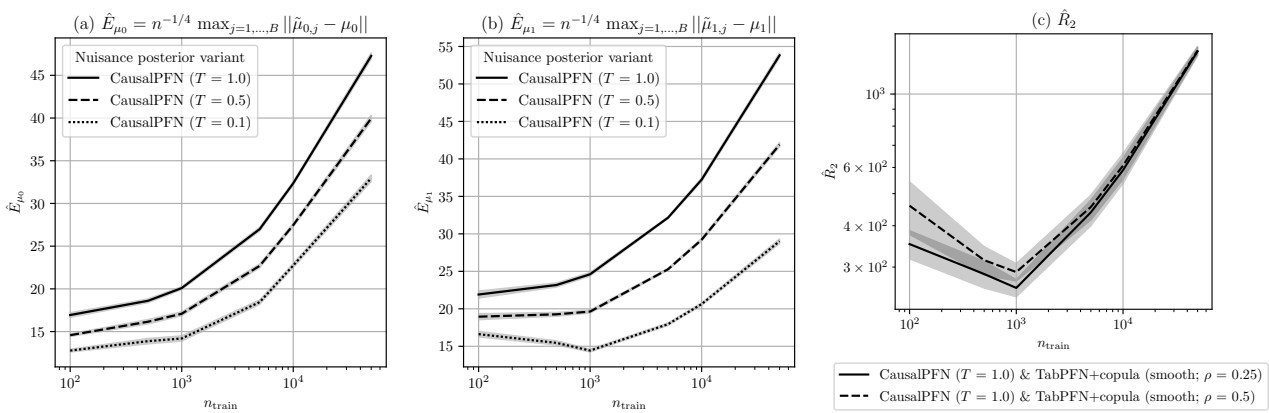

*Figure 9.* $L_2$-concentration check for every nuisance function (a)-(b) and for the second-order remainder $R_2$ (c) based on the synthetic data with varying size of the train data, $n_{\text{train}}$ (here $d_x = 25$). Reported: mean $L_2$ error $\hat{E}_\diamond / \hat{R}_2 \pm$ se over 10 runs (lower is better), where $\diamond$ corresponds to different nuisance functions: (a) $\diamond = \mu_0$, (b) $\diamond = \mu_1$ (note that propensity score is not available for CausalPFN).

### F.3. ATE Estimation

#### F.3.1. Sensitivity to Hyperparameters

Fig. 10 shows the quality of the (i) asymptotic ATE uncertainty wrt. varying hyperparameters $B$ (number of posterior draws) and $N$ (number of MP steps). Here, our MP-based ATE posteriors converge already when $B = 75$ and $N = 100$. Importantly, the performance of our MP-OSPC remains relatively stable under different hyperparameters, highlighting the first-order insensitivity to the misspecification of the nuisance functions posteriors.

Furthermore, Fig. 11 demonstrates the quality of the (i) asymptotic ATE uncertainty wrt. different copulas and their corresponding hyperparameters. Specifically, we report the results for two different copulas that are used for the outcome model: (a) a Gaussian copula with the correlation hyperparameter $\rho \in (0, 1)$, and (b) a Gumbel copula with the hyperparameter $\theta \geq 1$. Here, while the performance of the MP-plug-ins heavily depends on the choice of copulas and their hyperparameters $\rho/\theta$, the performance of our MP-OSPC remains stable, again highlighting the first-order insensitivity to the misspecification of the nuisance functions posteriors.

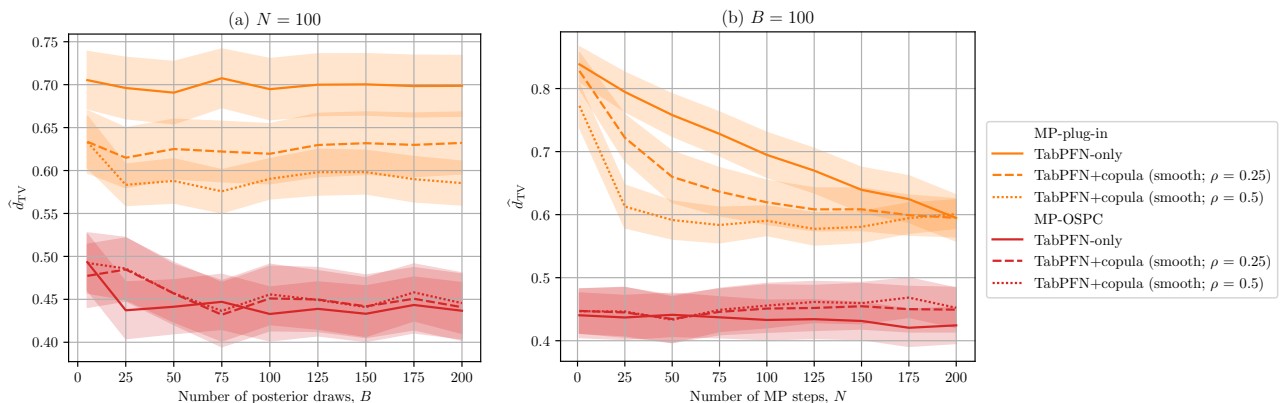

*Figure 10.* Quality of the asymptotic uncertainty for Bayesian ATE estimators based on the synthetic data with (a) varying number of posterior draws, $B$, and (b) varying number of MP steps, $N$ (here $n_{\text{train}} = 500, d_x = 25$). Reported: mean $\hat{d}_{\text{TV}} \pm$ se over 40 runs (lower is better).

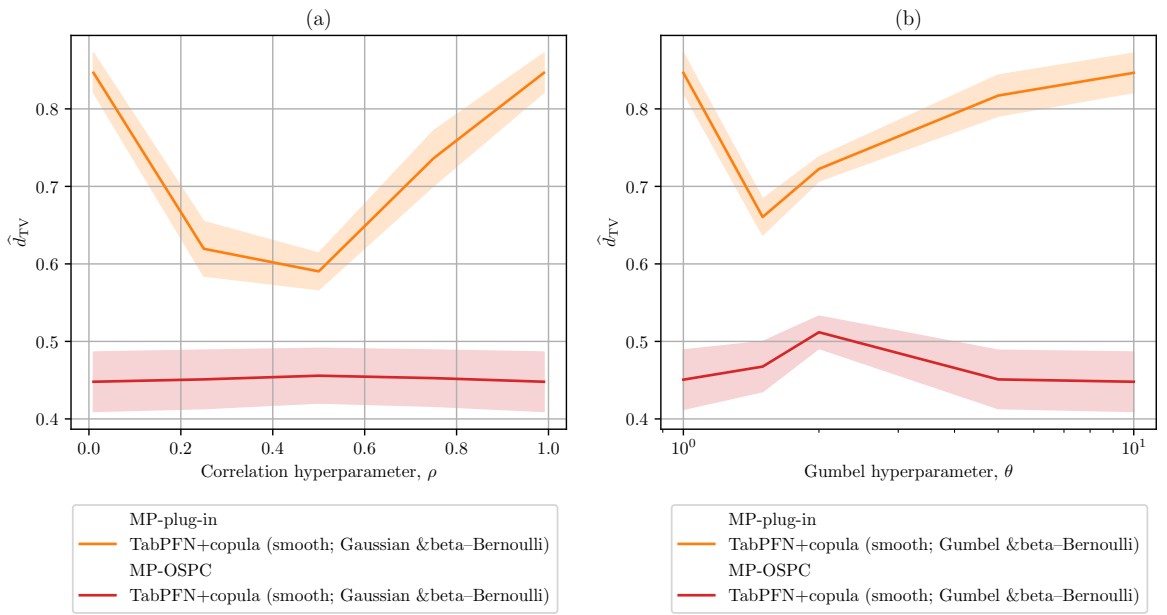

*Figure 11.* Quality of the asymptotic uncertainty for Bayesian ATE estimators based on the synthetic data with varying copula types, (a) Gaussian and (b) Gumbel, and corresponding hyperparameters, (a) $\rho \in (0, 1)$ and (b) $\theta \geq 1$ ($n_{\text{train}} = 500, d_x = 25, B = 100, N = 100$). Reported: mean $\hat{d}_{\text{TV}} \pm$ se over 40 runs (lower is better).

### F.3.2. ADDITIONAL SYNTHETIC DATASET RESULTS

Tables 3 and 4 report the (ii) finite-sample ATE estimation results with varying amounts of train data and covariate dimensionality, respectively. Here, our *MP-OSPC* always improves over naïve or MP-based plug-ins, as long as the data size is $n_{\text{train}} \leq 5000$ (which is expected as TabPFN struggles with consistent estimation of the nuisance functions for large $n$, as discovered in Sec. 6.1). Overall, we discovered that *good asymptotic performance of our* MP-OSPC *also translated into a well-calibrated finite sample uncertainty.*

| $\mu_a$ model | $\pi$ model | $n_{\text{train}}$ ATE estimator | 100 | 500 | 1000 | 5000 | 10000 |
|---|---|---|---|---|---|---|---|
| CausalPFN ($T = 1.0$) | — | Naïve plug-in | 0.965 | 1.000 | 1.000 | 1.000 | 1.000 |
| | TabPFN-only | MP-OSPC | **0.685** | 0.485 | 0.585 | 0.890 | 0.980 |
| | TabPFN+copula (smooth; $\rho = 0.25$) | MP-OSPC | 0.735 | 0.520 | 0.610 | 0.865 | 0.970 |
| | TabPFN+copula (smooth; $\rho = 0.5$) | MP-OSPC | 0.695 | 0.510 | 0.600 | 0.865 | 0.970 |
| CausalPFN ($T = 0.5$) | — | Naïve plug-in | 1.000 | 1.000 | 1.000 | 1.000 | 1.000 |
| CausalPFN ($T = 0.1$) | — | Naïve plug-in | 1.000 | 1.000 | 1.000 | 1.000 | 1.000 |
| TabPFN-only | — | MP-plug-in | 0.875 | 0.855 | 0.755 | 0.560 | 0.305 |
| | TabPFN-only | MP-OSPC | 0.865 | **0.340** | **0.330** | 0.820 | 0.870 |
| TabPFN+copula (smooth; $\rho = 0.25$) | — | MP-plug-in | 0.920 | 0.680 | 0.690 | 0.455 | **0.255** |
| | TabPFN+copula (smooth; $\rho = 0.25$) | MP-OSPC | 0.895 | 0.355 | 0.370 | 0.810 | 0.845 |
| TabPFN+copula (smooth; $\rho = 0.5$) | — | MP-plug-in | 0.920 | 0.590 | 0.550 | **0.390** | 0.335 |
| | TabPFN+copula (smooth; $\rho = 0.5$) | MP-OSPC | 0.860 | 0.350 | 0.340 | 0.825 | 0.830 |

Lower = better (best in **bold**, second best underlined)

*Table 3.* Quality of the (ii) finite-sample uncertainty for Bayesian ATE estimators based on the synthetic data with (a) varying size of the train data, $n_{\text{train}}$. Reported: mean $\widehat{d}_{\text{KS}}$ evaluated with 40 runs, grouped wrt. the underlying PFN. Here: $d_x = 25$.

| $\mu_a$ model | $\pi$ model | $d_x$ ATE estimator | 15 | 25 | 50 | 100 |
|---|---|---|---|---|---|---|
| CausalPFN ($T = 1.0$) | — | Naïve plug-in | 0.990 | 1.000 | 1.000 | 1.000 |
| | TabPFN-only | MP-OSPC | 0.500 | 0.485 | 0.560 | 0.575 |
| | TabPFN+copula (smooth; $\rho = 0.25$) | MP-OSPC | 0.540 | 0.520 | 0.575 | 0.585 |
| | TabPFN+copula (smooth; $\rho = 0.5$) | MP-OSPC | 0.555 | 0.510 | 0.560 | 0.585 |
| CausalPFN ($T = 0.5$) | — | Naïve plug-in | 1.000 | 1.000 | 1.000 | 1.000 |
| CausalPFN ($T = 0.1$) | — | Naïve plug-in | 1.000 | 1.000 | 1.000 | 1.000 |
| TabPFN-only | — | MP-plug-in | 0.955 | 0.855 | 0.880 | 0.690 |
| | TabPFN-only | MP-OSPC | **0.495** | **0.340** | 0.540 | **0.345** |
| TabPFN+copula (smooth; $\rho = 0.25$) | — | MP-plug-in | 0.760 | 0.680 | 0.820 | 0.695 |
| | TabPFN+copula (smooth; $\rho = 0.25$) | MP-OSPC | 0.570 | 0.355 | 0.535 | 0.375 |
| TabPFN+copula (smooth; $\rho = 0.5$) | — | MP-plug-in | 0.685 | 0.590 | 0.835 | 0.745 |
| | TabPFN+copula (smooth; $\rho = 0.5$) | MP-OSPC | 0.525 | 0.350 | **0.525** | 0.375 |

Lower = better (best in **bold**, second best underlined)

*Table 4.* Quality of the (ii) finite-sample uncertainty for Bayesian ATE estimators based on the synthetic data with (b) varying dimensionality of covariates, $d_x$. Reported: mean $\widehat{d}_{\text{KS}}$ evaluated with 40 runs, grouped wrt. the underlying PFN. Here: $n_{\text{train}} = 500$.

### F.3.3. ADDITIONAL IHDP RESULTS

In Table 5, we demonstrate the results in settings (i)–(ii) for the IHDP dataset. The IHDP dataset is known to have low overlap issues, and, thus, (i) asymptotic properties of the debiased estimators cannot be guaranteed (including our *MP-OSPC*). Still, *our MP-OSPC in combination with TabPFN achieves very good finite-sample calibration.*

| $\mu_a$ model | $\pi$ model | ATE estimator | (i) $\widehat{d}_{\mathrm{TV}}$ | (ii) $\widehat{d}_{\mathrm{KS}}$ |
|---|---|---|---|---|
| CausalPFN ($T = 1.0$) | — | Naïve plug-in | **0.239 ± 0.014** | 0.14 |
| | TabPFN-only | MP-OSPC | 0.319 ± 0.015 | 0.09 |
| | TabPFN+copula (smooth; $\rho = 0.25$) | MP-OSPC | 0.303 ± 0.013 | 0.13 |
| | TabPFN+copula (smooth; $\rho = 0.5$) | MP-OSPC | 0.311 ± 0.013 | 0.14 |
| CausalPFN ($T = 0.5$) | — | Naïve plug-in | 0.271 ± 0.015 | 0.10 |
| CausalPFN ($T = 0.1$) | — | Naïve plug-in | 0.359 ± 0.017 | 0.06 |
| TabPFN-only | — | MP-plug-in | 0.332 ± 0.019 | 0.08 |
| | TabPFN-only | MP-OSPC | 0.430 ± 0.021 | 0.09 |
| TabPFN+copula (smooth; $\rho = 0.25$) | — | MP-plug-in | 0.279 ± 0.014 | 0.08 |
| | TabPFN+copula (smooth; $\rho = 0.25$) | MP-OSPC | 0.271 ± 0.014 | 0.06 |
| TabPFN+copula (smooth; $\rho = 0.5$) | — | MP-plug-in | 0.281 ± 0.014 | **0.04** |
| | TabPFN+copula (smooth; $\rho = 0.5$) | MP-OSPC | 0.314 ± 0.014 | 0.07 |

Lower = better (best in **bold**, second best underlined)

*Table 5.* Quality of the (i) asymptotic and (ii) finite-sample uncertainties of ATE estimators based on the IHDP dataset. Reported: (i) mean $\widehat{d}_{\mathrm{TV}} \pm$ se over 100 splits, and (ii) mean $\widehat{d}_{\mathrm{KS}}$ evaluated with 100 splits. Both (i) and (ii) are grouped wrt. the underlying PFN.

### F.3.4. ADDITIONAL ACIC 2016 RESULTS

Fig. 12 (TabPFN-based estimators) and Fig. 13 (CausalPFN-based/combined estimators) show detailed (i) asymptotic and (ii) finite-sample ATE estimation results for each of 77 datasets from the ACIC 2016 datasets collection. Therein, we additionally provided information on the absolute degree of observed confounding for each dataset $|\Delta|$ and the median inverse propensity score. We observed that, for the majority of the datasets, $|\Delta|$ is below 0.25, which coincides with the priors for TabPFN/CausalPFN (see Fig. 2). Yet, *for datasets with $|\Delta| \gtrsim 0.25$, our* MP-OSPC *posterior almost always improved over naïve/MP-based plugins for both CausalPFN and TabPFN.*

### F.4. Memory Usage & Runtime

We report runtime and memory usage for synthetic datasets of increasing size in Fig. 14, also varying $B$ and $N$. There, TabPFN-only MPs are very GPU-memory/runtime intensive for an increasing number of posterior draws $B$ (as every draw has to be parallelized and requires $N$ subsequent TabPFN calls each with up to $O((n + N)^2)$ time complexity). At the same time, the copula-based MPs are more GPU-memory/runtime effective (as they require one initial TabPFN call with $O(n^2)$ time complexity and all the MP inference for $B$ draws can be easily vectorized).

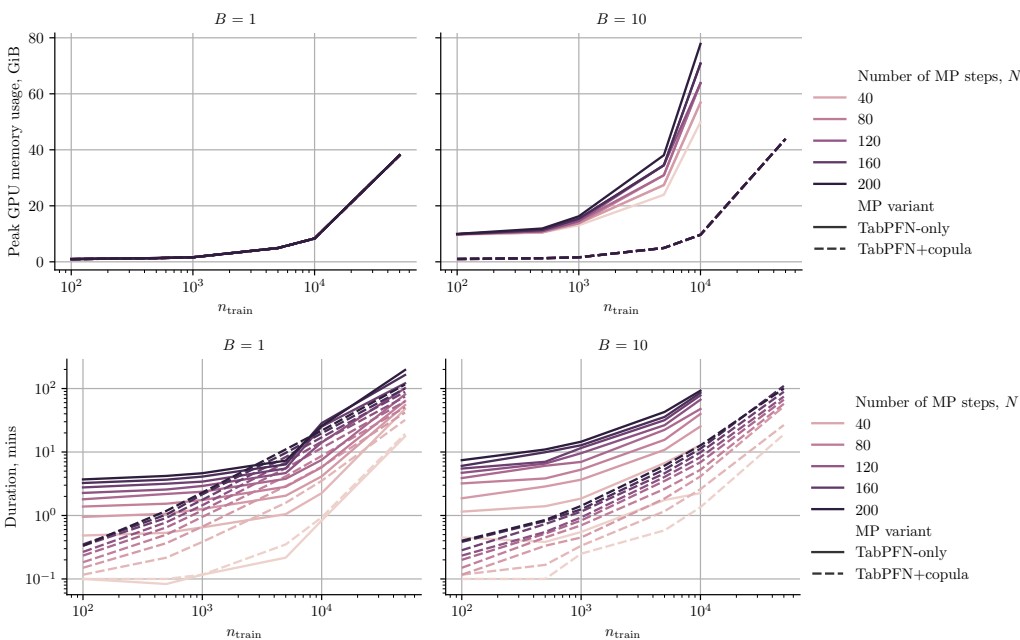

*Figure 14.* Peak GPU memory usage (top figure, in gigibytes (GiB)) and total runtime (bottom figure, in minutes) of ATE experiments for our *MP-OSPC* with different MP variants and varying number of MP steps based on the synthetic data with varying size of the train data, $n_{\text{train}}$ (here $d_x = 25$). Experiments are carried out on AMD EPYC 9455 48-Core CPU and 1 H200 Nvidia GPU.

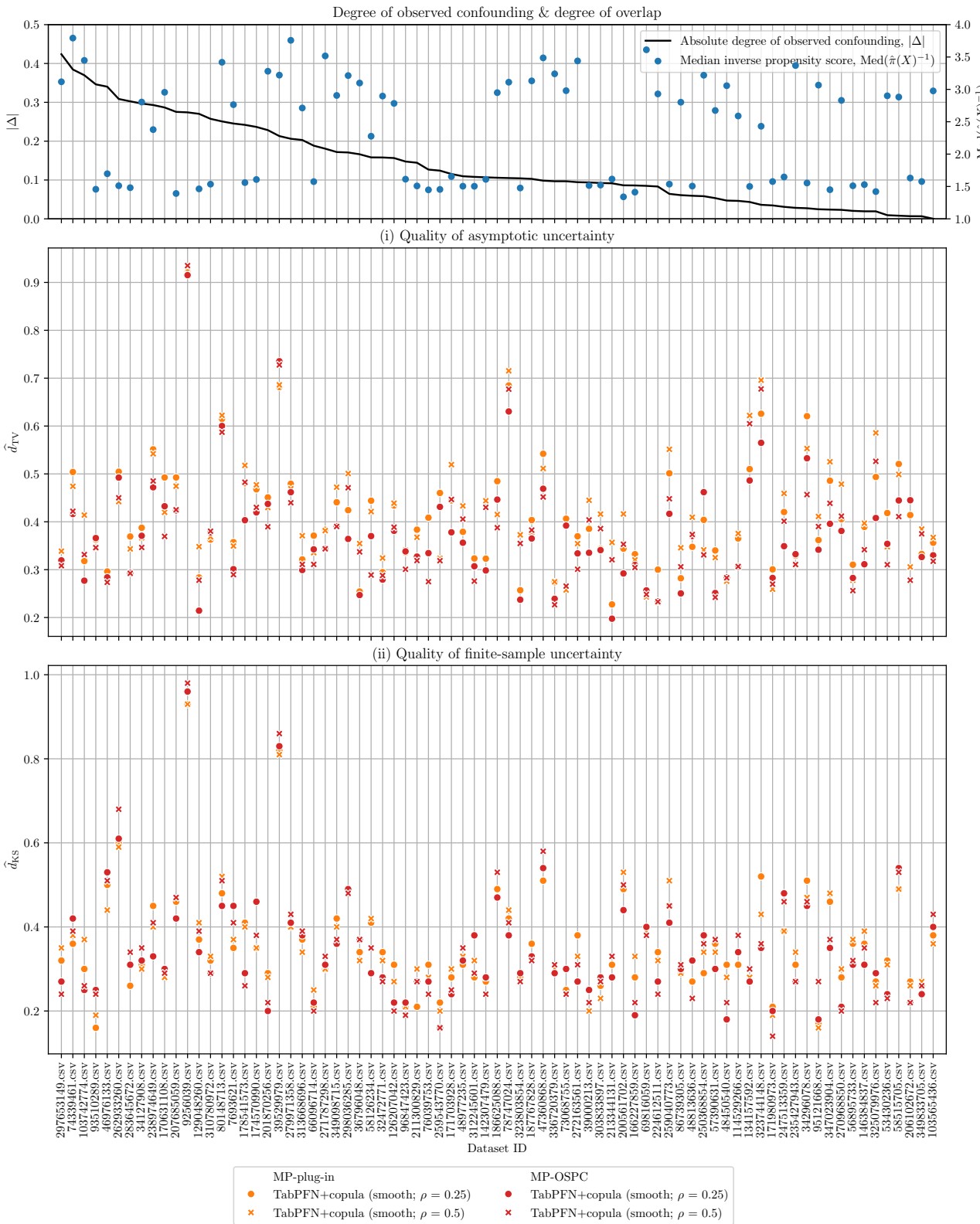

*Figure 12.* Full (i) asymptotic and (ii) finite-sample results of ATE estimation with TabPFN-based estimators for 77 semi-synthetic ACIC 2016 datasets. Datasets are sorted wrt. the absolute degree of measured confounding, $|\Delta|$; and the median inverse propensity score is provided for reference. Reported: (i) mean $\hat{d}_{\mathrm{TV}}$ and (ii) $\hat{d}_{\mathrm{KS}}$ over 10 runs (lower is better).

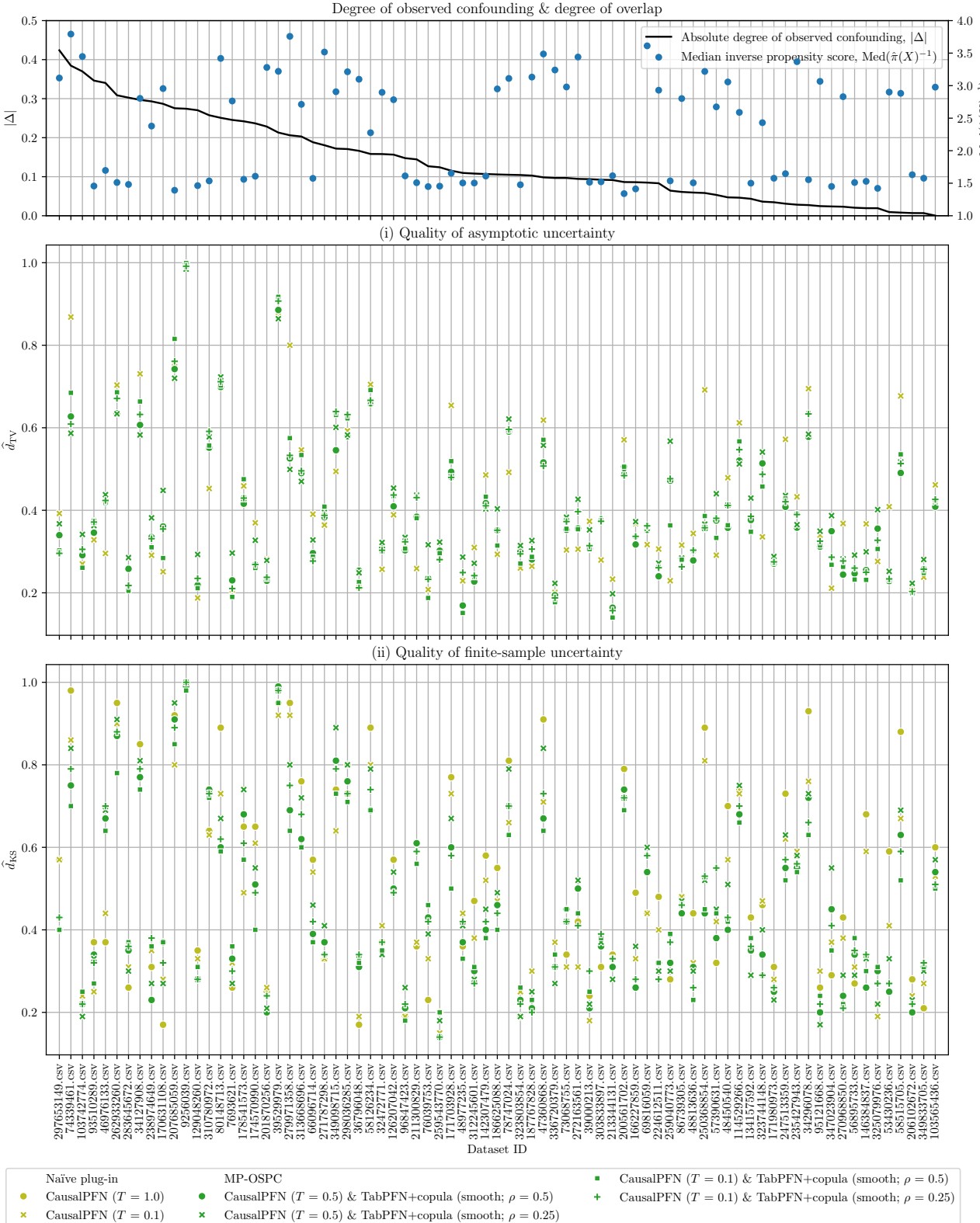

*Figure 13.* Full (i) asymptotic and (ii) finite-sample results of ATE estimation with CausalPFN-based/combined estimators for 77 semi-synthetic ACIC 2016 datasets. Datasets are sorted wrt. the absolute degree of measured confounding, $|\Delta|$; and the median inverse propensity score is provided for reference. Reported: (i) mean $\hat{d}_{TV}$ and (ii) $\hat{d}_{KS}$ over 10 runs (lower is better).

# G. Real-world Case Study

In the following, we provide a case study, where we apply our *MP-OSPC* to a real-world problem. Here, we want to study the effectiveness of lockdowns during the COVID-19 pandemic by using the observational data collected in the first half of 2020 (Banholzer et al., 2021). Specifically, we aim to estimate the average effect (ATE) of a strict lockdown on the incidence.

## G.1. Dataset

We use the multi-country dataset from Banholzer et al. (2021) and the pre-processing pipeline similar to Melnychuk et al. (2024).[7] The outcome $Y \in [-7, 0]$ is the weekly relative case growth on the log scale, defined as the ratio of new cases to cumulative cases. The treatment $A \in \{0, 1\}$ indicates whether a strict lockdown was implemented one week earlier. For the pre-treatment covariates $X$, we select three variables ($d_x = 4$): the relative case growth in the previous week, the relative case growth two weeks earlier, the relative case growth three weeks earlier, and the strict-lockdown indicator from two weeks earlier. We assume the observations are i.i.d. and that the causal assumptions (1)–(3) hold. We exclude observations with fewer than 20 cumulative cases. This leaves a total sample size of $n = n_0 + n_1 = 152 + 112$, corresponding to treated and untreated observations, respectively.

## G.2. Results

We show the results of ATE estimation in Fig. 15. Therein, we compared the uncertainties of frequentist and Bayesian ATE estimators based on different PFNs. Interestingly, while almost all estimators predict the negative ATE with high certainty/credibility (meaning a strict lockdown should reduce the incidence), the estimated ATEs differ among different PFNs: the location and spread vary for (a) TabPFN-based and (b) CausalPFN-based/combined estimators. *Importantly, the variants of our* MP-OSPC *Bayesian ATE estimator* match *the frequentist A-IPTW estimators the best, suggesting the best alignment between the two classes of estimators.*

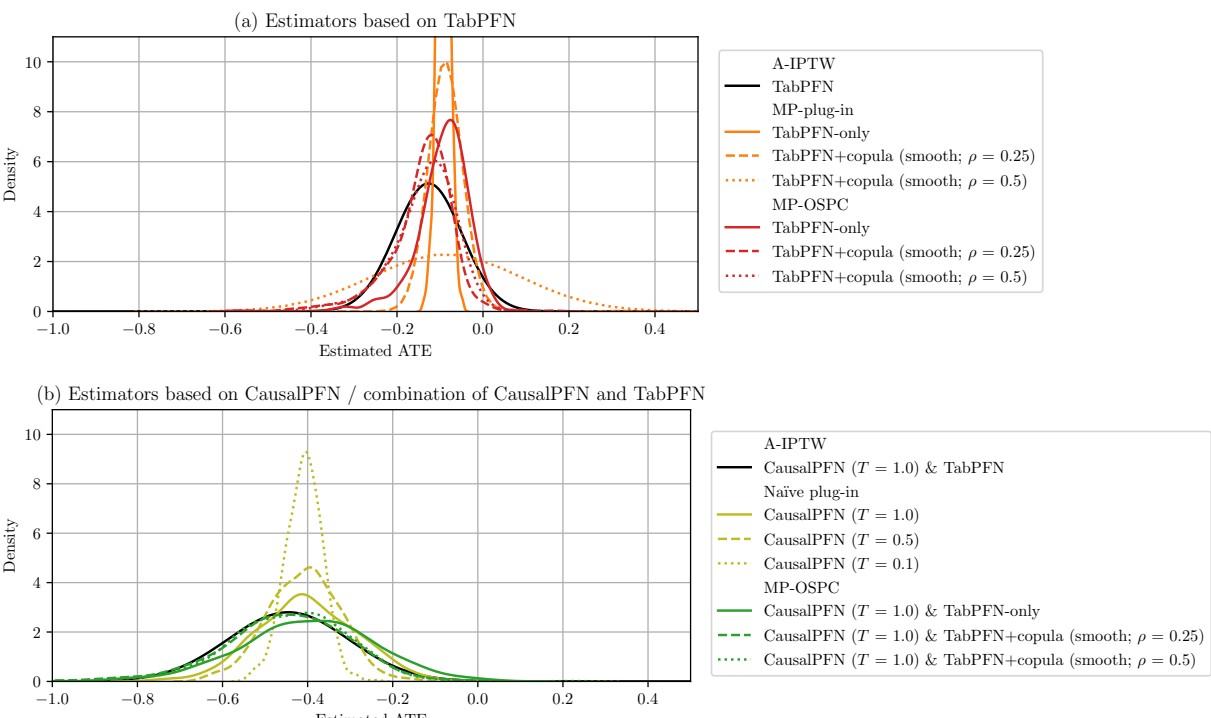

*Figure 15.* Uncertainty of frequentist and Bayesian ATE estimators based on different PFNs: (a) TabPFN-based and (b) CausalPFN-based/combined estimators. Reported: (i) density of an A-IPTW-based asymptotic normal distribution for frequentist estimators (in black); and (ii) posterior densities for Bayesian estimators (in colors). Both (i) and (ii) are based on 5-fold cross-fitting.

---

[7]The data is available at https://github.com/nbanho/npi_effectiveness_first_wave/blob/master/data/data_preprocessed.csv.

