# OpenReview forum: "Frequentist Consistency of Prior-Data Fitted Networks for Causal Inference"
_ICML.cc/2026/Conference — ICML 2026 regular_

### Official Review · Reviewer_ovZD · 2026-02-23

**Soundness:** 3
**Presentation:** 3
**Significance:** 4
**Originality:** 3
**Overall Recommendation:** 5
**Confidence:** 3

**Summary:**

This paper provides proofs and methodology for obtaining consistent average treatment effects (ATEs) from prior-data fitted networks (PFNs). The methodology they derive employs a martingale posterior approach to correct the bias that the usual causal PFNs employ. Part of the contributions of the paper includes an analysis of *when* the current PFNs would be viable for their approach.

**Compliance With Llm Reviewing Policy:**

Affirmed.

**Final Justification:**

I recommend acceptance. The authors have clarified my concerns about possible extensions to heterogeneous causal effects.

**Key Questions For Authors:**

1. Figure 5 and Table 2 indicate different uncertainty measures (namely, standard error and standard deviation), can the authors elaborate on this specific choice? Or was it a leftover from previous results that were not updated?
2. What prevents an immediate extension to heterogeneous effects? i.e., does this result naturally extend to CATE estimation? Or would the martingale correction not be necessary for accurate CATE estimation?
3. I'm having a hard time understanding if Step $N$ in Equations (13) and (14) only affects the 'Step $N$' in Equations (12) or if it would affect through all steps. Perhaps an explicit algorithm would help.

**Limitations:**

Yes

**Strengths And Weaknesses:**

## Soundness
The theoretical consistency (Theorem 1) hinges on the proof of Theorem 4 of Yiu et al (2025). It would be nice if the paper provided a full proof of said theorem, at least in the appendix.
## Presentation
The text in Figure 5 is really small, and hard to read in a printed version. I'd suggest to increase the font size. Besides that, the presentation is pristine and speaks of a well-polished paper. Notation is consistently used. The exposition covered the first questions that came to my mind, which I appreciate. The 'major' hiccup is that the current paper has no section for conclusions, as typically expected. However, being a (mostly) theoretical paper this is not a big problem, though it would be nice to hear from the authors about possible extensions to their contributions and other limitations that the current setup faces.

A minor issue is the overflowing of Eq. (14), that with some simplification of the notation (e.g., $\alpha_N= \alpha_N(v,{V'}^N$)) might be possible to fit within the margins.
## Significance
The results are significant, PFNs have grown in popularity for causal inference, having consistency properties is necessary for most causal inference
## Originality
I'm not up-to-date in PFNs and their theoretical properties, though I would be surprised (due to their recency) if this specific property had been considered previously.

---

> ### Author Rebuttal · Authors · 2026-03-30
>
> Thank you for a very positive evaluation of our work! We are grateful that you found our work well-polished and our results significant. Let us address your remaining concerns.
>
> ## Weaknesses
> - **Soundness**. Thank you for the suggestion! Given the limitations of the ICML rebuttal, we cannot provide the full text of the proof (only uploads of figures and tables are allowed). Below, we thus provide a sketch.
>
>    ---
>    Let $\hat\psi=\psi+P_n\\{\phi_\psi(Z;\eta)\\}$, i.e., the efficient A-IPTW limit. Consider the high-posterior-probability set $H_n$ from the theorem, with $\Pi(\tilde\eta\in H_n\mid D)\to 1$. It is therefore enough to prove the result conditional on $\tilde\eta\in H_n$.
>
>    Using the OSPC posterior, $\psi^{\mathrm{OSPC}}(\tilde\eta)\mid D = \psi(\tilde\eta)+BB_n \\{\phi_\psi(Z;\tilde\eta)\\}$, we decompose $\sqrt n\big(\psi^{\mathrm{OSPC}}(\tilde\eta)-\hat\psi\big) = \sqrt n(BB_n-P_n)\\{\phi_\psi(Z;\tilde\eta)\\} +G_n [\phi_\psi(Z;\tilde\eta) - \phi_\psi(Z;\eta)]-\sqrt n R_2(\eta,\tilde\eta)$, where $G_n=\sqrt n(P_n-P)$, and $R_2(\eta,\tilde\eta)$ is the second-order remainder. Now control the three terms.
>
>    First, for $\tilde\eta$, the Bayesian bootstrap CLT gives $\sqrt n(BB_n-P_n)\\{\phi_\psi(Z;\tilde\eta)\\} \rightsquigarrow N (0,Var[\phi_\psi(Z;\tilde\eta)])$.
>
>    Second, by the Donsker condition, $\sup_{\tilde\eta\in H_n} | G_n [\phi_\psi(Z;\tilde\eta)-\phi_\psi(Z;\eta) ] |=o_P(1)$.
>
>    Third, for the ATE the remainder satisfies $|R_2(\eta,\tilde\eta)| \lesssim \max_{a\in\{0,1\}} ||\tilde\mu_a-\mu_a|| \cdot ||\tilde\pi^{-1}-\pi^{-1}||$, hence Assumption (a) implies $\sqrt n\sup_{\tilde\eta\in H_n}|R_2(\eta,\tilde\eta)| \to 0$.
>
>    Finally, Assumption (a) also yields $\sup_{\tilde\eta\in H_n} ||\phi_\psi(\cdot;\tilde\eta) -\phi_\psi(\cdot;\eta)|| \to 0$, so $Var[\phi_\psi(Z;\tilde\eta)] \to Var[\phi_\psi(Z;\eta)]$.
>
>    Therefore, $\sqrt n\big(\psi^{\mathrm{OSPC}}(\tilde\eta)-\hat\psi\big)\mid \mathcal{D} \rightsquigarrow N (0,Var[\phi_\psi(Z;\eta)])$, which proves the semiparametric BvM theorem.
>
>    ---
>
>    **Action**. We will add the complete proof of the theorem to our revised manuscript.
>
> - **Presentation**. Thank you for your helpful suggestion.
>
>    **Action**. We will update the revised version of the paper as follows:
>    - (i) We increase the text size in Fig. 5.
>    - (ii) We add a proper conclusion section where we restated the limitations of our method and the main contributions.
>    - (iii) We fix the overflowing equations.
>
> ## Questions
> 1. **SE vs. STD**. True, both Figure 5 and Table 2 use different uncertainty estimates.
>
>    **Action**. In our revised manuscript, we will make all results consistent and only report SE.
>
> 2. **CATE extension**. Great question! The main obstacle for extending our work to CATE is that CATE, unlike ATE, is an _infinitely dimensional_ target estimand and, hence, the frequentist uncertainty is not well-defined. Specifically, the efficient influence function for CATE includes the Dirac-delta function, and, thus, a standard semi-parametric theory does _not_ apply in this case (and, therefore, the BvM property isn’t defined as well). One might consider Neyman-orthogonal CATE learners (e.g., DR-learner), yet they only yield point predictions and don’t provide asymptotic frequentist uncertainty for CATE itself.
>
>    **Action**. We will add a discussion of the challenges and potential strategies to extend our work to CATE.
>
> 3. **Step N**. We are sorry for the typo: Eq. 12 should have the following expression for step $N$: $N: Z'^{N} \sim \mathbb{P}(Z\mid \mathcal{D}'^{N-1}); \quad \mathcal{D}'^{N} =  \mathcal{D}'^{N-1} \cup \\{Z^{'N}\\}$. Therefore, step $N$ always depends on the results of step $N-1$.
>
>    In a nutshell, during step $N$, we are sampling a new datapoint $Z'^{N}$ from the previous PPD (based on the current dataset $\mathcal{D}'^{N-1}$) and appending it to the current dataset $\mathcal{D}'^{N-1}$. Eq. 13 and 14 then specify exactly how those updates are done for different variants of MPs: Eq. 13 samples $X'^{N} , A'^{N} $ from the empirical distribution and $Y'^{N} $ from the current PPD of a PFN, and Eq. 14  samples $X'^{N} , A'^{N} $ analogously and updates the PPD for $Y$ directly (w/o sampling).
>
>    **Action**. We added a more detailed algorithm to the revised version of our manuscript.
>
> Let us know whether we have clarified your concerns.

---

> > ### Author Rebuttal · Reviewer_ovZD · 2026-04-01
> >
> > My original concerns have been resolved, and the extensions to CATE seem promising!
> >
> > As a minor suggestion, when rereading the paper, the 'alarm' notation on Table 1 is not fully clear. Perhaps some explanation would be appropriate. Regardless, I will update my score accordingly.

---

### Official Review · Reviewer_vKEP · 2026-03-02

**Soundness:** 3
**Presentation:** 3
**Significance:** 3
**Originality:** 3
**Overall Recommendation:** 4
**Confidence:** 4

**Summary:**

This paper studies whether uncertainty quantification for Prior--Data Fitted Networks (PFNs) in causal inference yields frequentist-consistent posterior inference for the average treatment effect (ATE). The authors show that an implicit training prior in PFNs can induce a prior-induced confounding bias, so that a naive ATE posterior constructed from PFN pointwise posterior predictive distributions may not converge to the targeted frequentist asymptotic uncertainty. To address this issue, the paper proposes a one-step posterior correction (OSPC) based on the efficient influence function to calibrate the ATE posterior. Since implementing OSPC requires draws from a functional posterior over nuisance components, the paper uses martingale posteriors (MPs) to construct an approximate functional posterior from sequential PPD updates, and further develops a PFN--Copula variant for improved stability. Experiments on synthetic and semi-synthetic benchmarks indicate that the calibrated PFN posterior aligns more closely with the targeted frequentist asymptotic distribution and often improves finite-sample calibration.

**Compliance With Llm Reviewing Policy:**

Affirmed.

**Key Questions For Authors:**

(1) What is the motivation and intuition behind introducing a propensity-specific beta--Bernoulli mixture copula, and why is the standard Gaussian copula insufficient in this case?

(2) Given that Theorem 1 relies on $R_2 \to 0$, what evidence supports that the PFN+MP nuisance functional posterior satisfies the required contraction/product-rate conditions in practice (especially in light of Fig. 4)?

(3)How sensitive are the results to the choice of copula and key MP hyperparameters (e.g., $\rho$, $N$, and $B$)?

(4) What are the runtime/memory costs of MP-OSPC and how do they scale with $N$ and $B$ across different PFN backbones and datasets?

**Limitations:**

Yes

**Strengths And Weaknesses:**

Strengths:
(1) PFN-based ATE posteriors can inherit confounding bias from the synthetic pre-training prior and yield miscalibrated uncertainty. The paper uses semiparametric influence-function theory and a Bernstein–von Mises argument to motivate an OSPC correction that makes ATE uncertainty asymptotically match A-IPTW.

(2) Practical bridge from pointwise PPDs to functional posteriors: Recognizing that PFNs typically output only pointwise posterior predictive distributions rather than a functional posterior over nuisance components, the paper leverages Martingale Posteriors to construct an approximate functional posterior needed for one step posterior correction. It also proposes a PFN plus copula hybrid Martingale Posterior to improve robustness to martingale violations and to enhance practical usability.

(3) Thorough empirical validation with metrics matched to the claims: The evaluation spans synthetic and semi synthetic benchmarks such as IHDP and ACIC 2016, multiple PFN backbones including TabPFN, CausalPFN, and CausalFM, and several posterior construction variants including naive plug in, Martingale Posterior plug in, and Martingale Posterior one step posterior correction.

Weaknesses:
(1) The paper introduces a propensity-specific beta--Bernoulli mixture copula (in place of the Gaussian copula) as a key ingredient of the PFN+Copula MP pipeline, but provides limited explanation of why the propensity component requires a separate copula, what the construction is derived from (or how it should be interpreted), and how sensitive the results are to the copula form and its parameters. As a result, this component reads as relatively heuristic despite being central to the method.

(2) Theorem 1 establishes BvM-style alignment under nuisance functional posterior contraction and product-rate conditions (i.e., the second-order remainder satisfies $R_2 \to 0$). However, in the PFN setting, the paper notes that PFNs are only approximately Bayesian and that convergence rates for a given PFN are difficult to characterize. Moreover, the proxy check $\hat R_2$ (Fig. 4) is non-monotone and shows rebound behavior at larger $n_{\text{train}}$.

(3) Although MP is motivated as potentially expensive and TabPFN-only MP is excluded on ACIC-2016 due to runtime, the paper does not provide systematic runtime/memory measurements or scaling curves with respect to the number of MP steps $N$ and posterior draws $B$.

(4) Key MP hyperparameters (e.g., $N$ and $B$) are largely fixed, and the exploration of dependence strength $\rho$ is limited. More importantly, the paper provides few direct diagnostics of whether the sequential updates satisfy (even approximately) the martingale property, and it lacks coverage/calibration diagnostics tracked along MP iterations.

---

> ### Author Rebuttal · Authors · 2026-03-30
>
> We are grateful for your positive review of our paper! We are glad that you found our method sensible and sound. In the following, we address the weaknesses and questions you raised.
>
> ## Weaknesses and Questions
> - **W1 & Q1**. Great question! We used a beta-Bernoulli mixture copula because it is specifically designed for posteriors involving **binary** random variables [1] (e.g., in our case, the propensity score is a conditional probability of the binary treatment). By contrast, Gaussian copulas are suited to (absolutely) **continuous** random variables, such as the continuous outcome in our setting. Hence, the copula choice is largely fixed by the variable type: outcome model <-> Gaussian copula, treatment model <-> beta-Bernoulli mixture copula. The main tunable hyperparameter in both cases is the correlation coefficient $\rho$ (discussed in **W4 & Q3**).
>
>    **Action**. We will add this explanation to the revised paper.
>
> - **W2 & Q2**. This point is one of our main findings: **the underlying PFN**, rather than MP-OSPC, is the main obstacle to achieving the BvM result (we kindly refer to **W2 & Q2** of **Rev. F85Z**).
>
>    It is not surprising that existing PFNs do not fully satisfy BvM conditions: PFNs are inherently empirical and are not guaranteed to remain consistent beyond the dataset sizes seen during training (e.g., TabPFN v2.5 was trained up to $n=50000$). We confirm this empirically in Fig. 4, where TabPFN begins to break down already for $n>5000$. To explain this, we additionally analyzed the error of each nuisance function separately (-> (a) https://tinyurl.com/4p56cxfn) and found that the main failure comes from the posterior over inverse propensity scores. These posteriors stop concentrating already for $n>1000$, exactly where performance begins to deteriorate in Fig. 4, because propensity scores may approach 0 or 1. This suggests an important future direction: PFNs specifically tailored to inverse propensity score modeling (= Riesz regression [2]).
>
>    We still believe PFNs+MP-OSPC can be useful in practice, especially in **medium-data regimes**, much like TabPFN in standard non-causal settings. For example, in ACIC-2016 ($n=4802$), TabPFN+MP-OSPC performs well on the majority of datasets. In practice, we therefore advise using PFNs+MP-OSPC when (1) the dataset size lies within the PFN training range and (2) the inverse propensity posterior is concentrated.
>
>    **Action**. We will add these practical implications to the revised paper.
>
> - **W3 & Q4**. Great idea! Our MP-OSPC can only be combined with TabPFN, since it is the only PFN that provides a **full PPD** for the **observed data** (see Table 1).
>
>    A simple approximation of the time and memory complexity of MP-OSPC is as follows ($W$ = number of PFN weights, $B$ = number of posterior draws, $N$ = number of MP steps):
>    - PFN-only: $O(B (N+n)^2 W)$
>    - Copula-based MPs: $O(n^2 W + B (N+n) n)$
>
>    We report runtime and memory usage for synthetic datasets of increasing size in (b) https://tinyurl.com/4p56cxfn, also varying $N$ and $B$. As $B$ increases, runtime and memory for PFN-only MP-OSPC grow dramatically, whereas they stay relatively stable for copula-based MPs.
>
>   **Action**. We will add this runtime analysis to the revised paper.
>
> - **W4 & Q3**. Great suggestions!
>    - **Martingale property**. As shown in [3], the martingale property does not hold for TabPFN but does hold for copulas. Still, since the martingale property is sufficient but not necessary for MP convergence [4], we conducted new empirical diagnostics. We recorded the average posterior standard deviation over $X$ for different nuisance functions recovered with MPs ((c) https://tinyurl.com/4p56cxfn). With a few exceptions for large $\rho$, the posterior standard deviation flattens for all the nuisance functions as the number of MP steps increases (e.g., around $N=300$ for $n=100$, $N=250$ for $n=500$), indicating convergence to a limiting distribution. Importantly, even though it takes up to $N = 300$ for the full MP convergence (in some cases even more), our MP-OSPC allows for good ATE estimation already for $N=100$ (as we demonstrate in the following).
>
>    - **Sensitivity to choice of copula and MP hyperparameters**. We also ran additional synthetic ATE experiments varying $\rho$, $B$, and $N$ ((d) https://tinyurl.com/4p56cxfn). We found that MP-OSPC is relatively insensitive to these hyperparameters, unlike MP-plug-ins. This supports its debiasedness: uncertainty in nuisance posteriors affects ATE uncertainty only through higher-order terms.
>
>    **Action**. We added new experiments with diagnostics and sensitivity checks to the revised paper.
>
> We hope to have addressed all of your concerns!
> ## Refs:
> - [1] https://arxiv.org/abs/2103.15671
> - [2] https://arxiv.org/abs/2104.14737
> - [3] https://arxiv.org/abs/2505.11325v2
> - [4] https://arxiv.org/abs/2510.25154

---

> > ### Author Rebuttal · Reviewer_vKEP · 2026-04-03
> >
> > The rebuttal still does not fully address my core concerns. In particular, the authors effectively acknowledge that current PFNs do not systematically satisfy the theoretical conditions in practice, which weakens the theorem’s support for the actual method. Moreover, the copula choice is still justified mainly by variable-type compatibility rather than empirical necessity, while the complexity, sensitivity, and scope of applicability remain insufficiently clarified. This is especially concerning if the method is effectively limited to TabPFN, in which case the claims should be stated more cautiously and with a narrower scope.

---

> > > ### Author Response · Authors · 2026-04-05
> > >
> > > Thanks for your swift response! In the following, we want to address your remaining concerns.
> > >
> > > ### Drawbacks of current PFNs
> > > We understand your worries regarding our final method. In the pursuit of the idealized BvM property, we faced the fundamental issue of PFNs: they cannot guarantee a consistent estimation of nuisance functions in both frequentist and Bayesian senses. Arguably, we cannot fix this inconsistency simply with a posterior correction method without re-training or fine-tuning the underlying PFN.
> > >
> > > Thus, we agree that our final method does not provide a foolproof solution for Bayesian ATE estimation. However, our work should rather be considered as a **cautionary tale** and provides an **important foundation for future work**:
> > > - Given the rising popularity of PFNs, we discover an important limitation of their naive application (i.e., as plug-in estimators).
> > > - We demonstrate an important gap in existing PFNs (e.g., there is no PFN to directly model the inverse propensity scores).
> > >
> > > => We will clarify our limitations in the Discussion section of the paper.
> > >
> > > ### Copula choice
> > > Indeed, in our work, we used the default choices of copulas (provided in the seminal paper on martingale posteriors [1]). Interestingly, we haven’t found any works on the MPs that used anything other than Gaussian/beta–Bernoulli copulas for real-valued/binary variables, respectively (correct us if we are wrong).  Still, this default choice turned out to be an effective choice across multiple semi-synthetic datasets. On a conceptual level, our method is agnostic to the choice of a copula and even to the choice of the martingale posterior method as a whole (e.g., PFNs-only MPs do not use copulas). Still, as we saw with the synthetic data, the inherent inconsistency of the underlying PFNs is the main obstacle to the overall method and **not** the specification of the MP variant, copula variant, or copula hyperparameters. This is consistent with prior work on parametric MPs, where the overall consistency **mainly depends on the consistency of the initial PPD** [2].
> > >
> > > => We performed additional synthetic ATE experiments, where we used a Gumbel copula with varying hyperparameter $\theta \ge 1$ as an outcome MP model (-> see Fig. 5 in the **updated PDF (d)** https://tinyurl.com/4p56cxfn). Again, we saw that while the MP-plug-ins are highly sensitive to the choice of copula and its hyperparameters, the MP-OSPC performs similarly well for all the values of $\theta$ and different variants of copulas (Gumbel vs. Gaussian).
> > >
> > >
> > > ### Limitation to the TabPFN
> > > True, out of all the existing PFNs, only TabPFN yields PPDs for both the outcome model and propensity model. However, the OSPC applies to the combination of models: e.g., CausalPFN can be used to naively model $\mu_a$ posteriors (i.e, in the $x$-independent mode), and TabPFN + our MP wrapper provides the posterior for $\pi$. In this regard, the OSPC framework is fully model-agnostic, where every nuisance function can be modelled with an arbitrary Bayesian model (this is analogous to the frequentist A-IPTW estimators).
> > >
> > > => Upon your question, we added an OSPC ATE posterior based on the combination of CausalPFN and TabPFN (as described above) as one of our baselines.  Based on the synthetic data results (-> see the **updated PDF (e) & (f)**  https://tinyurl.com/4p56cxfn), we saw the following:
> > > - e. *$L_2$ convergence check*. As seen from the second-order remainder $\hat{R}_2$, the divergence of the CausalPFN-based posteriors for $\mu_a$ can be partially compensated with a convergence TabPFN-based propensity score MP posterior. This again highlights the double robustness property of our MP-OSPC.
> > > - f. **ATE estimation**. Our MP-OSPC (= CausalPFN &TabPFN combination) improves over naive CausalPFN-based plug-in estimators.
> > >
> > > We hope to have addressed your remaining concern, and we appreciate your effort in reviewing our paper!
> > >
> > > ## References
> > > - [1] https://arxiv.org/abs/2103.15671
> > > - [2] https://arxiv.org/abs/2410.17692

---

### Official Review · Reviewer_jTPm · 2026-03-10

**Soundness:** 3
**Presentation:** 3
**Significance:** 3
**Originality:** 3
**Overall Recommendation:** 4
**Confidence:** 4

**Summary:**

The paper analyzes frequentist consistency of PFNs for ATEs. In particular the main contributions are: (i) Diagnosing challenges achieving posterior contraction at $\sqrt{n}$-rate (i.e., BvM consistency), when using PFNs to construct Bayesian plug-in estimators of ATEs - which the authors claim arise due to prior-induced confounding bias and more generally first-order sensitivity of the plugin posterior to nuisance function uncertainty), (ii) Proposing a Bayesian AIPW estimator (OSPC correction) using nuisance functions derived from a PFN, and proving BvM consistency under technical conditions, and (iii) adapting Martingale Posterior (MP) sampling techniques to turn predictive posteriors $p(y|D,a,x)$, $p(a|D,x)$ into posteriors on the nuisance functions $\pi,\mu$, so-as to implement the proposed estimator.  The proposed method appears to outperform naive application of PFNs for ATEs via plug-in estimators in experiments.

**Compliance With Llm Reviewing Policy:**

Affirmed.

**Final Justification:**

The authors propose (to my knowledge) the first work for using PFNs to construct Bayesian debiased estimators for average effects, that can achieve frequentist (BvM) consistency. This is a useful contribution, but I initially had concerns regarding how the paper positions the problem being solved (i.e., the fundamental problem being PICB vs. general slow plug-in rates).

The authors' rebuttal has clarified my understanding of the problem, and I am broadly happy with the changes they have proposed to make in the revision to make the discussion of these problems clearer. As such, I have raised my initial score. The remaining limitations concern the scope of the paper's analysis (i.e., there is no formal analysis of PICB or conditions under which PFN+MCP constructions satisfy the conditions required for BvM consistency), but I don't think addressing these is necessary to meet the bar for publication.

**Key Questions For Authors:**

I would appreciate it if you could address my queries and concerns above regarding:
1. The scope of what Figure 2. shows re. prior-induced bias / concentration on unconfounded problems.
2. How this bias specifically prevents BvM consistency (rather than say, the issue being generally due to too-slow contraction rates).
3. How the prior-induced bias may induced under-estimation of ATE uncertainty
4. Where the prior causal datasets $D^c$ originate from (I assume the PFN training distribution?).

**Limitations:**

The main limitations of the paper I think worth discussing are:

1. The paper does not verify that PFN+MP posterior constructions satisfy the required properties, or discuss conditions under which this may be true - verifying this is challenging, but worth discussing.

2. The PFN prior-bias issue is empirical/heuristic rather than formal - thus it is not guaranteed to correctly explain the main reason for lack of frequentist consistency.

**Strengths And Weaknesses:**

**Soundness**

The overall methodology/aims are well-motivated and the use of OSPC to bias correct PFN-based ATE estimators is mathematically principled. However, I have a few concerns about the paper’s justification of its main narrative and one issue with how the theoretical consistency argument is presented.

- **Prior Induced Confounding Bias:** The narrative argued by the authors that "*the prior induced confounding bias is, therefore, a main obstacle for the frequentist consistency of the PFNs*", I do not find sufficiently justified:

1. The authors claim that the confounding proxy $\triangle$ is concentrated around zero in Figure 2 in all cases. However, while Figure 2 does show that $\triangle$ is *centered* around zero (which is reasonable to expect when averaging over many training datasets), it is unclear how (tightly) concentrated this is. For example, for TabPFN $d_x > 5$, there is significant mass outside the [-0.5,0.5] interval. Since the natural scale of $Y$ is not given here, it is difficult to determine what values of $\triangle$ are large or small.

I suspect the authors are trying to imply that, whatever the scale/support, mass should be more evenly distributed across the values of $\triangle$ if many datasets with large degrees of confounding were used, but I think the support/scale does affect the truth of this statement/reasoning (for a fixed scale of $y$).

2. The main issue to achieving the BvM consistency in Definition 1 appears to be more generic: namely, that non-parametric posteriors are generally unable to obtain the required faster-than-$\sqrt{n}$ rate of posterior contraction. This, as far as I am aware, does not relate directly to the so-called bias shown in Figure 2, which is for a fixed $n$. More generally, BvM consistency may fail regardless of whether the so-called prior-confounding bias is present. If so, the paper positioning of *"bias-induced inconsistency"* does not seem entirely correct.

3. Relatedly, $\triangle$ appears to be computed by over prior causal datasets (from the PFN training distribution?)), rather than on learned posteriors given observed datasets, so it is not obvious how much this translates into PFN posteriors which are concentrated on nearly unconfounded models.

4. It is also not clear to me how an under-representation of training datasets which are strongly confounded means the ATE uncertainty may be under-estimated, as claimed before the Takeaway on pg 5.

As a more minor point, the authors also state the phenomenon worsens as $d_x$ increases, but for TabPFN the phenomenon actually improves with increasing $d_x$.

- **Theorem 1 Proof**: the proof of Theorem 1 is outsourced to Thm 4 of Yiu et al (2025). However in that paper, the proof only seems to cover the Donsker case, while in Thm1 in the present paper allows for sample splitting to be used instead. While the sample-splitting route is fairly standard for semi-parametric efficiency results, the proof is not correct as is and either the theorem/proof statement should be modified accordingly.

**Presentation**

The paper is generally well-structured and clearly written, however the language used in places is slightly confusing:

- The paper refers to the BvM consistency Definition 1 as "*"frequentist consistency"*. I think this is a bit of a misnomer because it actually defines a CLT-type property rather than a LLN-type property. For instance, for the Bayesian plug-in estimator, we may still have posterior consistency:
$$\Pi_{\eta \mid \mathcal D_n}(\|\psi(\eta) - \psi_{true}\| >\epsilon \|) \overset{n\to \infty}{\to_p} 0, \quad \forall \epsilon > 0$$
which is arguably a better definition for "frequentist consistency" in this setting. This creates some confusion when the paper argues that PFN-based plug-in estimators cannot achieve frequentist consistency.

- The paper could make a bit clearer that the prior-bias suffered by PFNs is diagnosed empirically / via an example, rather than formally.

- It is not clear whether the prior causal datasets $D^c$ are those used to train the original PFN, or separate synthetic datasets drawn in some other way. More information on this would be helpful.

**Significance**

As PFNs are becoming increasingly popular and important tools, developing a framework for using them to construct Bayesian ATE estimators with good frequentist estimators is an important problem, which the authors address. As a result, the paper makes clearly relevant contributions.

**Novelty**

The main techniques used in the paper (OSPC, MP) come from existing work either on Bayesian ATE inference or PFN posterior sampling techniques. However the present work appears to be the first to analyze the use of PFNs in constructing Bayesian ATE estimators, and is also the first to lay out a general framework for combining PFNs with these techniques (OSPC, MP) to construct such estimators with good frequentist properties. These aspects are in my view the main novelties of the paper.

**Overall**

At present I have narrowly leaned to reject, due to my concerns around the main narrative. However I invite the authors to clarify the aspects I have raised and I do think the proposed method (OSPC+MP for PFN-based ATE estimation) is a worthwhile contribution.

---

> ### Author Rebuttal · Authors · 2026-03-30
>
> Thank you for a very rigorous and detailed review. We are happy that you recognised our contribution and found the method novel and relevant. Below, we address your concerns.
>
> ## Weaknesses
> ### Soundness
> - **PICB**. Thank you for raising this issue; our justification for OSPC via PICB can indeed be strengthened.
>    1. You are right that Δ in Fig. 2 depends on the scale of $Y$, making its magnitude hard to interpret. => We thus re-evaluated the priors using standardized $Y[a]$ (-> (a) https://tinyurl.com/52a3t8x4) and corrected a swap between CausalPFN and CausalFM priors in the original submission. The same pattern remains: the prior for Δ is highly concentrated for TabPFN/CausalPFN (Δ $\in [-.3,.3]$), while for CausalFM it concentrates with increasing $d_x$ (Δ $\in [-.5,.5]$). To judge if these values are large, we found out that Δ $\approx -.5$ for synthetic data, $\approx -.02$ for IHDP, and $\in [-.4,.4]$ for ACIC 2016. On ACIC 2016, MP-OSPC is especially effective when |Δ|>.25 (-> (b) https://tinyurl.com/52a3t8x4).
>
>    2. We agree that the main obstacle for BvM of plug-in posteriors is the slow concentration of Bayesian nonparametric posteriors (and this is different from PICB). Yet, our point is that, in Bayesian ATE inference, this issue is further worsened by PICB: when the prior for Δ is concentrated near zero, the plug-in posterior becomes over-regularized in the wrong direction, biasing the ATE plug-in posteriors toward the difference in means [1]. Also, upon your question, we found that |Δ| upper-bounds the magnitude of the OSPC debiasing term (we kindly refer to **W1 & Q1** of **Rev. F85Z**). Hence, even if some PFNs implicitly perform debiasing like TMLE estimators (e.g., CausalPFN/CausalFM may arguably perform debiasing out of the box as they were trained on the causal datasets), they should fail when Δ is large, as we saw with synthetic data. => We will thus revise Sec. 5.1 to emphasize both issues equally: slow concentration and PICB.
>
>    3. Yes, Δ was computed from the prior causal datasets used to train PFNs: for CausalPFN/CausalFM, these are directly available in code or from the authors; for TabPFN, we additionally sampled treatment assignments in the same spirit as its synthetic classification data generation. We added the code for Fig. 2 to the anon repository. To answer your main question, the exact value of PICB in plug-in posteriors is tractable only in simple parametric models [2]. In our PFN setting, the prior is given by a complex NN-based sampling scheme, so PICB is generally intractable. Still, semiparametric Bayesian theory [3,4] implies that the posterior can only concentrate within the prior support; thus, if Δ is large, no amount of data can fully correct the plug-in ATE posterior for **any Bayesian estimator**. => We will clarify this in the revision.
>
>    4. Thank you for spotting the ambiguity. => We will remove the phrase “underestimating the uncertainty of ATE” and replace it with the more precise term “asymptotic bias induced by the misspecified prior” (= PICB).
>
>    - **Minor point**. Indeed. => We will clarify that l. 251-252 refers specifically to GPs. For TabPFN/CausalPFN, the concentration of Δ is the same wrt. $d_x$.
> - **Theorem 1**. True: Sample splitting is not fully equivalent to the Donsker condition. It must still be combined with suitable $\sup$-convergence and envelope conditions (Ass. 1(c)* in [5]). => We will correct the theorem statement accordingly.
> ### Presentation
> - **Consistency**. We agree that there are two notions of consistency. In our paper, we used “frequentist consistency” in the BvM sense, following e.g. [6]. => We will make this explicit.
> - **PICB**. You are right that, for PFNs, this is empirical in the sense that the prior for Δ is obtained by sampling. However, as noted above, no plug-in posterior can concentrate beyond the prior support. => We will clarify this.
> - **Prior causal datasets**. Yes, we ensured that we used the same datasets as in PFN training (-> **PICB** (3)).
> ## Questions
> 1. -> **PICB** (1) & (3).
> 2. In short, PICB further over-regularizes plug-in posteriors toward models with small confounding (-> **PICB** (2)).
> 3. -> **PICB** (4).
> 4. -> **PICB** (3).
> ## Limitations
> 1. We agree that current PFNs lack formal asymptotic guarantees and, thus, are mainly useful for medium-sized datasets. Importantly, the key obstacle to BvM lies in the PFNs themselves rather than in our MP-OSPC wrapper (we kindly refer to **W2&Q2** of **Rev. F85Z**).
> 2. We agree, but emphasize that it is an **important discovery**, relevant for future work on causal PFNs.
>
> => We will add a limitations section to summarize these points.
>
> Given a strict character limit, let us know if we answered your questions.
> ## Refs:
> - [1] https://arxiv.org/abs/2111.03897
> - [2] https://arxiv.org/abs/1602.02176
> - [3] https://arxiv.org/abs/math/0607023
> - [4] https://doi.org/10.1214/12-EJS675
> - [5] https://arxiv.org/abs/2410.17692
> - [6] https://arxiv.org/abs/1705.03439

---

> > ### Author Rebuttal · Reviewer_jTPm · 2026-04-03
> >
> > Thank you for addressing my points in detail. I still have a couple of things I'm trying to understand / get to the bottom of.
> >
> >  **Q1. PICB Concentration:** Thank you for presenting scaled analogues of PFN densities. Should I interpret the plotted densities in Fig (a) as for variables with mean 0 and variance 1? Essentially, it still isn't clear to me how we are quantifying concentration here. For instance, one could plot two distributions and say one is more concentrated near zero than another if one has much more mass near a neighborhood of zero. What is the natural reference here?
> >
> > **Q2. What is the exact effect of PICB on frequentist consistency?**  This is still the main point I am trying to understand.
> >
> > In the paper, the narrative strongly suggests that PICB is the central obstacle:
> > - *“The prior-induced confounding bias is, therefore, a main obstacle for the frequentist consistency of the PFNs.”*
> > - *“To circumvent the prior-induced confounding bias, we suggest employing a one-step posterior correction (OSPC).”*
> >
> > However, in the rebuttal you now say that the main obstacle for BvM validity of plug-in posteriors is the slow concentration of Bayesian nonparametric posteriors, and that PICB is an additional effect that worsens this. These are quite different positions.
> >
> > Could you therefore state as explicitly as possible what PICB changes mathematically? For example:
> > - Does PICB affect the **rate** at which the plug-in posterior converges?
> > - Is it an **asymptotic obstruction**, or a **finite-sample regularization effect**?
> >
> > A concrete way to phrase my confusion is: suppose a PFN-based plug-in posterior already fails Definition 1 purely because of the generic slow-rate semiparametric issue. In that case, what additional failure mode is introduced by PICB, and what aspect of that failure is resolved by OSPC?
> >
> > **Q3. Concentration versus support.**
> > In the rebuttal you write that *“the posterior can only concentrate within the prior support; thus, if $\Delta$ is large, no amount of data can fully correct the plug-in ATE posterior for any Bayesian estimator.”*
> >
> > This is true, but my understanding was that the concern was about concentration rather than support. Is there any evidence of the support of $\Delta$ being mis-specified?
> >
> >
> > **Summary**
> >
> > My reading of the paper is that it is (to my knowledge) the first work to address the challenge of Bayesian debiased inference with PFNs. I think this is a useful contribution. However, the paper positions the problem as due to PICB, when it seems like it is (mostly?) a Bayesian analogue of the usual slow-rate problem of plug-in estimators. Your rebuttal has said you will revise to give equal weight to the two problems, but I'm still not sure yet what specific effect PICB has that OSPC solves.
> >
> > I think it is crucial the paper gets this positioning right to avoid readers coming to the wrong conclusion about what problem is being solved. If you could address my questions above that would be very helpful.
> >
> > **Update post reply-rebuttal**
> >
> > I thank the authors for clarifying my points. The answers have broadly satisfied by concerns about the narrative and so I raise my score by 1. I would please request that the authors do the following for the revision:
> >
> > - Overlay some standard distributions (if too hard to fit in the main text then at least in the appendix) as reference points for densities of $\Delta$ (with the same mean and variance), so one can assess how concentrated mass is relative to these (e.g., one could pick a Gaussian or triangular distribution). If the effect is quite stark, this would strengthen the arguments in the main text.
> >
> > - Add the details of the PICB vs. Slow rate discussion to the main text, so that readers are clear on how these problems interact and what the additional effect of PICB is.

---

> > > ### Author Response · Authors · 2026-04-04
> > >
> > > We are grateful for your follow-up questions and the ability to clarify our work and correct our previous claims.
> > >
> > > - **Q1**. Your understanding is correct:
> > >    > Should I interpret ... with mean 0 and variance 1?
> > >
> > >    The new Fig. 2 demonstrates the concentration of the prior for Δ on a **standard deviation scale**. For example, both TabPFN and CausalPFN are concentrated between [-0.25, 0.25], meaning that the amount of measured confounding is at most a quarter of one standard deviation.
> > >    > What is the natural reference here?
> > >
> > >    To the best of our knowledge, there isn’t a natural, well-agreed unit to measure confounding for a continuous outcome $Y$ (e.g., see different variants in Table 1 [1]): For some datasets, even tiny amounts of confounding may flip the sign of the ATE, and, thus, they can be considered significant (thus, there isn’t a simple natural reference). In our work, we chose a difference between the ATE and the difference in means – a default, simple-to-calculate variant to quantify the measured confounding [1, 2].
> > >
> > > - **Q2**. Those are important questions!
> > >    > Does PICB affect the rate at which the plug-in posterior converges?
> > >
> > >   It does for general semi-parametric Bayesian models (i.e., when the target parameter is finite-dimensional, but the model includes infinite-dimensional nuisance functions), as mentioned on page 5 of [3]. Namely, the oversmoothing of the nuisance functions may induce a bias in the plug-in posterior distribution of the target parameters.  If this bias is vanishing with n, we say it contributes to the convergence rate of the plug-in posterior.
> > >    > Is it an asymptotic obstruction, or a finite-sample regularization effect?
> > >
> > >    Whether the PICB vanishes asymptotically with growing n (and, thus, contributes to the convergence rate of the plug-in posterior) depends on _how the nuisance functions are modelled_ and whether _the likelihood dominates over the nuisance functions prior_:
> > >    - Given a very general, non-smooth functional prior (e.g., Dirichlet),  the likelihood does not dominate over the prior, and the bias may be constant with increasing n [4]. This leads to a general inconsistency (= the notion you mentioned in the initial review) of the posteriors. Therefore, in this case (= infinite-dimensional non-parametric priors), the PICB is distinct from the convergence rate of the plug-in posterior.
> > >    - Given smooth functional priors (e.g., GPs), the likelihood often dominates over the prior, and, thus, the bias vanishes with increasing n, yet generally slower than $\sqrt{n}$ [3]. For example, this is the case for [5], where the PICB decreases for non-degenerate DGPs. In this case, indeed, the PICB is simply a contributing factor for the slow concentration rates.
> > >
> > >    For the sake of generality (and the fact that the PICB is present specifically for Bayesian inference but not for the frequentist), we decided to distinguish two concepts: the PICB and slow concentration rates. Yet, given the similarity of the PFNs with the Gaussian processes, we acknowledge that, in our setting, the PICB can be simply a contributing factor for the slow concentration rates.
> > >
> > >   > what aspect of that failure is resolved by OSPC?
> > >
> > >    Given that, the right assumptions are satisfied (i.e., of Theorem 1), the OSPC corrects for either type of PICB (both non-vanishing and vanishing). => We will clarify these technical details in Sec. 5.1 & Sec. 5.2.
> > >
> > > - **Q3**. Thank you for bringing this up! Indeed, in our initial rebuttal, we make a (stronger) statement about the support of the prior distribution. However, as we figured out in the answer to Q2 above, the PICB also exists when no limitations on support of Δ are made (e.g,  when GPs are used).
> > >    > Is there any evidence of the support of being mis-specified?
> > >
> > >    In the case of PFNs, the honest theoretical answer is that we don’t know (as not all the details about the prior are described in the original publication). Yet, given the empirical nature of PFNs, both statements about the support and the concentration can be arguably used interchangeably (even if in rare cases some prior synthetic datasets have a large Δ, it barely changes the fit of a PFN). => To be very precise, we will only refer to the prior concentration and won’t assume anything about the support of Δ.
> > >
> > > Thank you for your thorough review of our paper! Your comments and suggestions are really helpful to sharpen the motivation and the narrative for our method.
> > >
> > > ## Refs:
> > > - [1] https://arxiv.org/pdf/2405.08738v1
> > > - [2] https://biostats.bepress.com/ucbbiostat/paper341
> > > - [3] https://arxiv.org/pdf/1808.04246
> > > - [4] https://doi.org/10.1214/aos/1176349830
> > > - [5] https://arxiv.org/pdf/1602.02176
> > >
> > > ## Response to update post reply-rebuttal
> > > We are happy to hear that we clarified your concerns, and we are grateful for raising your score! We will surely revise our paper to add the requested changes.

---

### Official Review · Reviewer_F85Z · 2026-03-16

**Soundness:** 3
**Presentation:** 4
**Significance:** 3
**Originality:** 3
**Overall Recommendation:** 5
**Confidence:** 2

**Summary:**

The paper discusses the use of prior-data fitting networks (PFNs) for causal inference tasks, such as estimating the Average Treatment Effect (ATE). According to this approach, we train a neural network with synthetic data, generated by different random processes (such as Gaussian), and use it to predict values of interest for real-world data. This approach allows us to have access to the Posterior Predictive Density (PPD) of a quantity of interest (ATE, propensity score, etc) and, thus, quantify its uncertainty.

However, the authors argue that the standard PFNs used so far mis-quantify this uncertainty because the synthetic data on which they are trained do not involve high-degree confounding. They support this claim with experimental results for the three PFNS; TabPFN, CausalPFN, CausalFM.

The authors argue that for a PFN estimator to have desirable behavior, it should be consistent with the classical frequentist estimator, i.e., the standard estimators that are based on the estimation of certain nuisance functions. In the case of ATE studied here, the authors compare to the performance of AIPTW estimator. They propose a method called *one-step posterior correction*, which uses both the standard PFN estimator for ATE based on the outcome regression function, as well as the correction term used in AIPTW, based on the propensity scores. They prove a theorem that posits the success of their method under standard convergence assumptions of the nuisance parameters.

Finally, they propose a practical implementation of their method based on combining the known PFNs with martingale posteriors. They implement this method on synthetic and semi-synthetic data and present the improvement of their method compared to standard approaches.

**Compliance With Llm Reviewing Policy:**

Affirmed.

**Final Justification:**

I suggest the paper is accepted. I believe the paper addresses and resolves an important open problem in the use of PFNs for causal estimation. During the rebuttal the authors addressed my comments thoroughly and provided intuition for their motivation, approach and validity of their technique.

**Key Questions For Authors:**

1. Do you believe you could argue more formally about the insufficiency of previous PFNs methods (without the correction step)?

2. Do you think there is a way to improve MP-OSCP so that it satisfies Theorem 1 (at least empirically)? Why does the L2 error become larger with the sample size? Do you have any intuition about it?

**Limitations:**

yes

**Strengths And Weaknesses:**

Strengths:
1. The paper is very nicely written and well-structured, making reading interesting.
2. The authors point out an important open problem in the use of PFNs for causal estimation that should be addressed. They propose a solution (*one-step posterior correction*) that is intuitive and compares well with the well-known, doubly-robust ATE estimator AIPTW.

Weaknesses:
1. The observation that the standard methods based on PFNs suffer from confounding bias is solely based on experimental results, without theoretical support for this behaviour.
2. The final proposed MP-OSCP method does not satisfy the convergence assumption necessary for Theorem 1 to apply. In fact, it doesn't even *empirically* satisfy it, since the experimental convergence check in Figure 4 seems to fail.

---

> ### Author Rebuttal · Authors · 2026-03-30
>
> Thank you for your positive review of our work! We are grateful that you found the paper nicely written and well-structured, and acknowledged the importance of our method. Below, we elaborate on the mentioned weaknesses and questions.
> ## Weaknesses and Questions
> - **W1 & Q1**. Thank you for bringing attention to this issue!
>
>    Indeed, we established only _empirically_ that PFNs are prone to prior-induced confounding bias (PICB), since they are trained on synthetic datasets with a limited amount of measured confounding (Fig. 2). To prove this _theoretically_, one would need an analytic form of the PFN training priors and then push them forward into the space of Δ. In practice, this is intractable because the synthetic training distributions are defined through complex sampling schemes rather than closed-form priors.
>
>    More broadly, PFNs are an inherently empirical approach, and their theoretical analysis remains largely open. One possible direction is to exploit their similarity to GPs [1], which are known to suffer from PICB. Moreover, when synthetic causal datasets are constructed without explicitly correcting for PICB, it is not surprising that this bias appears in existing PFNs.
>
>    In summary, our motivation for debiasing PFN-based plug-in estimators rests on two main points:
>    1. **Slow concentration rate**. PFN-based plug-in posteriors inherit the concentration behavior of the underlying PFN. As discussed in [2], Bayesian methods cannot generally concentrate faster than $1/\sqrt{n}$. Therefore, PFN-based plug-in ATE posteriors will typically concentrate more slowly than comparable frequentist A-IPTW estimators, so frequentist consistency need not hold in general. While some causal PFNs may perform partial debiasing out of the box, they still face the second issue below.
>    2. **PICB**. In Bayesian ATE estimation, slow concentration is further aggravated by PICB: if the prior amount of measured confounding is concentrated around zero, the posterior for $\mu_a$ becomes over-regularized, pushing the plug-in ATE toward the difference in means. In addition, prompted by your question, we found an important connection between PICB and OSPC: the PICB approximately upper-bounds the magnitude of the OSPC debiasing term,
>
>       $|\mathbb{E}(\phi_\psi(Z;\tilde\eta))|\lesssim|\mathbb{E}\bigg(\frac{A-\pi(X)}{\pi(X)(1-\pi(X))}(Y-\mu_A)\bigg)|=|\Delta|,$
>
>       where $\mu_A=\mathbb{E}(Y \mid A)$ and $\tilde\eta$ is taken as the ground-truth propensity score $\pi$. Hence, even causal PFNs trained on synthetic causal datasets cannot yield properly debiased plug-in posteriors when |Δ| is large.
>
>    => To streamline our motivation for debiasing PFNs, we added these arguments to the revised paper.
>
> - **W2 & Q2**. You raise an important question. We have several reasons to believe that the convergence failure of PFN-based ATE estimators is mainly due to the **underlying PFN itself**, rather than to our MP-OSPC wrapper.
>    1. The approximation error of MP-OSPC can be reduced by increasing the number of MP steps $N$. Its convergence in $N$ was studied in [3], where the concentration speed is roughly $O(1/\sqrt{N+n})$. Moreover, [4] shows that BvM for parametric MPs mainly requires consistency of the initial posterior predictive distribution, which in our case is supplied by the PFN.
>    2. In Fig. 2, we showed empirically that the second-order remainder $\hat{R}_2$ starts increasing already for $n>5000$, suggesting inconsistency of the PFN itself beyond the maximum training sizes of the prior datasets. We also conducted additional experiments ((a)&(b) https://tinyurl.com/ceayjfth):
>       a. Decomposing $\hat{R}_2$ shows that PFNs already struggle to model inverse propensity score posteriors for n>1000, precisely where performance begins to deteriorate in Fig. 4. This is plausible because propensity scores can approach 0 or 1, making inverse propensity modeling difficult.
>       b. We evaluated variants of MP-OSPC with n=500 and different N, and performance already plateaued around N=100.
>
>    Thus, we view **the PFN itself**, not MP-OSPC, as the main obstacle to achieving the BvM result. Still, there are ways to fix it, such as (1) to train PFN on larger dataset sizes and (2) to adapt PFN specifically for modelling inverse propensity scores. If the initial PPD came from a proper, consistent Bayesian method, as suggested in [5], then the BvM result could in principle hold exactly. Nevertheless, debiased PFN-based estimators can still be useful in **medium-data regimes**, similar to TabPFN in standard non-causal settings.
>
>    => We will add this insight, the new ablations, and a broader discussion of practical challenges for achieving the BvM result to the revised paper.
>
> We hope to have answered all of your questions!
> ## Refs
> - [1] https://arxiv.org/abs/2112.10510
> - [2] https://doi.org/10.1017/9781139029834
> - [3] https://arxiv.org/abs/2505.11325v1
> - [4] https://arxiv.org/abs/2410.17692
> - [5] https://arxiv.org/abs/2103.15671

---

> > ### Author Rebuttal · Reviewer_F85Z · 2026-04-02
> >
> > Thank you for your detailed explanations. My questions are resolved and I'll raise my score accordingly.

---

### Decision · Program_Chairs · 2026-04-30

**Decision:**

Accept (regular)

**Comment:**

This paper looks at the issue that PFN based causal estimators can be biased by their prior in the training data. I think this was a crucial issue for which the authors proposed a convincing fix. While I have still doubts on the practicality of the general approach of PFNs, this is a clear improvement over the prior iteration of methods.